# High Arctic channel incision modulated by climate change and the emergence of polygonal ground

Shawn M. Chartrand [1,2] ✉, A. Mark Jellinek[2], Antero Kukko [3],
Anna Grau Galofre[4], Gordon R. Osinski[5] & Shannon Hibbard[5,6]

Stream networks in Arctic and high-elevation regions underlain by frozen ground (i.e., permafrost) are expanding and developing in response to accelerating global warming, and intensifying summertime climate variability. The underlying processes governing landscape dissection in these environments are varied, complex and challenging to unravel due to air-temperature-regulated feedbacks and shifts to new erosional regimes as climate change progresses. Here we use multiple sources of environmental information and physical models to reconstruct and understand a 60-year history of landscape-scale channelization and evolution of the Muskox Valley, Axel Heiberg Island. A time series of air photographs indicates that freeze-thaw-related polygon fields can form rapidly, over decadal time scales. Supporting numerical simulations show that the presence of polygons can control how surface runoff is routed through the landscape, exerting a basic control on channelization, which is sensitive to the timing, duration and magnitude of hydrograph events, as well as seasonal air temperature trends. These results collectively highlight that the occurrence and dynamics of polygon fields modulate channel network establishment in permafrost-rich settings undergoing changes related to a warming climate.

Global warming and the intensity of warming variability have been amplified in the Arctic[1] since the 1980s[2,3]. These features of climate change are manifest through a recent acceleration in the seasonal frequency of new thermokarst features[4–6], and the spatial expansion of stream networks[7–11]. In temperate climates, channels incise the landscape and form stream networks that expand by erosion of river bed and bank sediments related to surface runoff[12–15] and groundwater sapping[16,17]. Contributions to these water flows include rainfall, seasonal snowmelt and glacial melt in seasonally cold and mountainous regions. In Arctic and high-elevation permafrost environments, contributions to surface and groundwater also include seasonal inputs from thawing ground ice and permafrost[6–10,18–21]. However, how these

hydrologic processes interact to influence channel development through erosion of frozen or partially frozen sediment particle substrates is unclear[18]. In particular, summer surface water runoff is modulated by snowmelt, glacier and ice cap melt, intermittent lake outburst floods, and melt water from seasonal pore ice in active layer soils and from permafrost[6–11,18–21]. These hydrologic sources deliver water to the surface over distinct temporal and spatial scales that change during single melting seasons[6–11,18–21]. Furthermore, the relative water contributions may change as active layer soil depths increase with rising seasonal mean temperatures[9,19,20], and in relation to the intensity of warming variability for climate change. As a result, we generally understand the sources of surface waters within permafrost

[1]School of Environmental Science, Simon Fraser University, Burnaby, BC V5A 1S6, Canada. [2]Department of Earth, Ocean and Atmospheric Sciences, University of British Columbia, Vancouver, BC V6T 1Z4, Canada. [3]Department of Remote Sensing and Photogrammetry, Finnish Geospatial Research Institute, National Land Survey of Finland, Espoo 02150, Finland. [4]Laboratoire de Planétologie et Géosciences, CNRS UMR 6112, Nantes Université, Le Mans Université et l'Université d'Angers, Nantes 44322, France. [5]Department of Earth Sciences, University of Western Ontario, London, ON N6A 5B7, Canada. [6]Jet Propulsion Laboratory, California Institute of Technology, Pasadena, CA 91011, USA. ✉e-mail: shawn_chartrand@sfu.ca

landscapes, but not how these sources collectively govern the timing and pace of channelization and landscape change[11,18].

The flow pathways and erosive properties of meltwater sources are affected by the character and permeability of periglacial soils[9,19,20], and the dynamic growth and evolution of polygon fields[7,8,11,22,23] (hereafter referred to simply as polygons) related to processes including thermal contraction, ice wedging[11,24] and ice lens growth[25,26]. The occurrence of active layer detachment slides and retrogressive thaw slumps of soil-ice-meltwater mixtures also influences hydrologic flow pathways by modifying the landscape topographically, and by providing point sources of potentially erosive meltwater flows together with fine and coarse sediments[9,19]. As a result, and in marked contrast to temperate settings, channel initiation and evolution in Arctic and high-elevation environments depend on how the seasonal production and flow of meltwaters are influenced by periglacial landforms and the spatial distribution of seasonal ground ice and relatively long-lived permafrost[21]. In particular, the strength and erodibility of the river bed and bank sediments by surface flows depend on seasonally varying volume fractions of pore ice, which can bind active layer soils depending on the thermal state of the material[18].

Here we characterize the consequences of these effects for stream network development and growth over an approximately decadal time scale. We explore whether and how periglacial landforms influence channelization at the basin scale, noting that the overall process is complex, due to the influence of numerous co-evolving environmental factors[27]. Key remaining knowledge gaps are the underlying mechanics of sediment particle erosion under frozen, or partially frozen conditions[18], as well as the relative contributions to erosion and

channel development from seasonally varying water sources[7–11,18,19,21]. Indeed, increased rates of channel bed erosion may contribute to a positive climate feedback by enhancing carbon release through increasing the availability of thawed organic soil carbon for microbial decomposition[6,28]. Consequently, and in light of accelerated Arctic warming (Fig. 1), an acute knowledge gap at the intersection of climate change science and geomorphology is a clearer understanding of channel initiation and stream network development in permafrost environments[18,27]. Here we combine air photographs from 1959 with field observations and LiDAR data collected in 2019 to characterize a remarkable 60-year landscape evolution in the Muskox Valley, Axel Heiberg Island. Details of this landscape change history are further explored using physical models, which combined highlight basic controls over stream network development, including how lake outburst floods can influence channel erosion and deposition, mediated through polygons and permafrost melting responses. We integrate our findings and propose a framework for describing landscape change, and to help better understand underlying processes governing the channelization responses of Arctic and high-elevation permafrost landscapes, as a result of a warming climate marked by intensifying summertime variability.

## Results and discussion
### Channel-polygon interactions from field observations and a physical model
We conducted fieldwork during the summer of 2019 within the Qikiqtani Region of Axel Heiberg Island, NU, located in the Canadian Arctic Archipelago (Fig. 1) ("Methods"). Our primary field site in

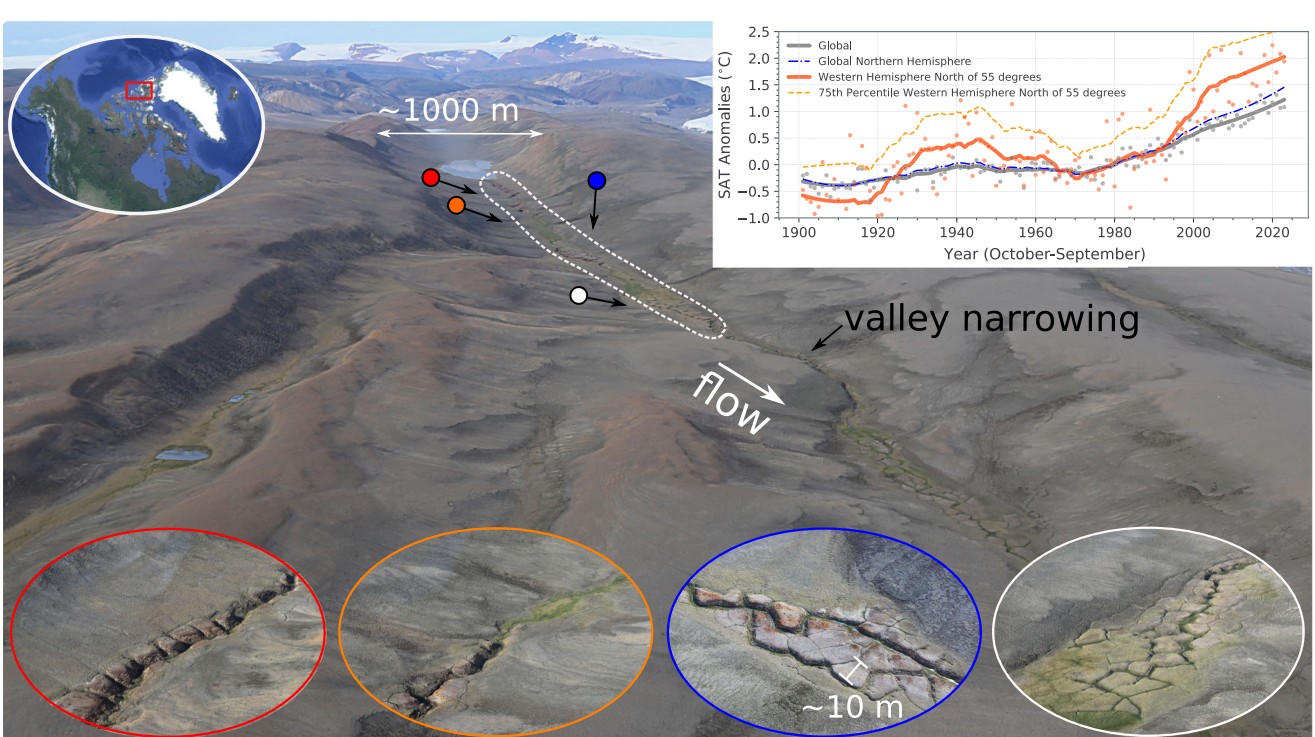

**Fig. 1 | Muskox Valley. Oblique aerial photograph of Muskox Valley showing a polygon-controlled stream channel.** The full valley is 8 km long and 1 km wide and has a flattened U-shape. Oblique image taken by authors looking up the valley toward a lake near the drainage divide. The dashed white outlined area shows the approximate limits of high-resolution three-dimensional LiDAR data collected in July 2019 ("Methods"). Oblique image center coordinates: 8872800, 457600, UTM projection, Zone 16. Inset oblique low-altitude images illustrate channelization characteristics of the valley floor. The inset images were taken by the authors and are color-keyed with arrows pointing to the location and showing the general camera orientation of the photographs. The noted polygon length scale in the

second-from-left inset image is consistent between the four images. The inset plot in the upper right shows average land surface air temperature (SAT) anomalies in degrees Celsius for the entire globe, the northern hemisphere and in the western hemisphere for latitudes from 55 to 90°N, and longitudes from −165 to 0°, including the 75th percentile SAT. CRUTEM5 alternative grid dataset, www.cru.uea.ac.uk/cru/data/temperature[42]. Trend lines are calculated using the Savgol Filter, SciPy library, with window size of 25 and a polynomial order of 2. Satellite image sources for upper left inset image: Esri | Maxar (DigitalGlobe) | GeoEye | Earthstar Geographics | CNES/Airbus DS | USDA | USGS | AeroGRID | IGN | IGP | | AEX | Getmapping | swisstopo | GIS User Community, Imagery ID: 10df2279f9684e4a9f6a7f08febac2a9.

Muskox Valley offers a unique opportunity to examine landscape change and channel network development from the early stages of response within an environment characterized by spatially continuous permafrost and a valley shaped by past glaciation[29]. Muskox Valley is situated east of the Müller Ice Cap and has a polar desert climate with an annual mean air temperature of −19 °C ("Methods"). The relatively cold regional climate implies that widespread thaw and melt within this arid environment could be a recent response to climate change, and its amplification in the Arctic over the past several decades[2–11] (Fig. 1), as well as an important driver of basin channelization.

The current expression of surface channels along Muskox Valley is diverse, depending on the presence or absence of polygons. Low-altitude photographs and our high-resolution LiDAR-based digital elevation model (DEM) show that the character and extent of erosional channel incision are spatially discontinuous (Figs. 1 and 2). Whereas well-developed channels are common around the edges of large polygons and within interconnected troughs[11], channelization is diffuse or non-existent where relatively poorly developed polygons occur in wetlands. Consistent with previous observations[7,8,11,22,23,30], we find that polygon geometry can influence and, in some places, govern the channel path or paths both across and down topographic gradients at the valley scale. Hydrodynamic modeling using the LiDAR-based DEM of a roughly 400 m segment of Muskox Valley ("Methods") shows that where polygons are present, water runoff for steady flow conditions is organized along interconnected polygon troughs, with larger locally averaged downstream velocities compared to wetland regions (Fig. 3 and Supplementary Fig. 1). These observations and results suggest topographic guiding of surface flows within polygon troughs enhances the likelihood of erosion there (Fig. 3 and Supplementary Table 1), and possibly increases the seasonal or event-driven rate of erosion. Furthermore, the spatial and discrete probability distribution of surface flows is strongly dependent on flow magnitude, and the presence or absence of polygons (Fig. 3 and Supplementary Fig. 1).

Valley floor topographic cross-sections from our LiDAR-based DEM provide insightful context for the differing characters of polygon-channel interactions in Muskox Valley (Fig. 2a). In the uppermost part of the basin at Station 200 m, the topographic cross-section is characterized by a V-shaped channel set within a polygon field. Downstream at Station 720 m, the cross-section is relatively flat across the valley bottom. Farther down the valley at Station 1120 m, there are two channels that occur within the troughs of polygons. The occurrence of multiple channels at this location is one example of a strong local control on the path of water runoff by polygons[7,8,22,23,30] (Fig. 3). At Station 1390 m, by contrast, the valley bottom is flat and flanked by raised polygons to the north. Runoff at this location has a tendency to spread across the valley floor, which is characterized by low topographic variability, and relatively high surface roughness (Fig. 3).

Observations from the 1959 photographic records (discussed in more detail below with Fig. 4) and our LiDAR-based DEM from 2019 show that sediment redistribution by fluvial erosion and deposition is actively reshaping the upper basin topographic profile. Fluvial transport is presently influenced by approximately isotropic and periodic polygon fields with a characteristic scale of ~10 m that is independent of the local valley slope (Fig. 2a) ("Methods"). Channelized and unchannelized segments of the valley floor are distinguished neither by the predominant topographic character of the polygon fields nor on the basis of either the absolute profile slope or its changes down-valley (Fig. 2b). This slope independence of the channelization response is surprising because the local profile steepness for channelized and unchannelized segments are similar (1–8%), which would conventionally be interpreted to mean that erosion potential by flowing water should be broadly equivalent[31,32]. In marked contrast to conventional views of the main controls over channel incision in mid-latitude temperate regions, the spatial pattern of channelization in Muskox Valley is arguably independent of channel steepness and grain size distributions[31], raising the importance of addressing critical knowledge gaps related to the erosion mechanics of river bed and bank sediments in permafrost settings[18,27].

A further conventional test for incision processes driven primarily by the erosive power of surface flows is the valley Width-Depth (W/D) ratio ("Methods"). Erosion and hence channelization is more likely within locations of smaller ratios because narrower valley profiles cause surface flows to converge into comparatively deeper and faster flows with relatively high erosive power (e.g., Fig. 1–location of valley narrowing). Width-Depth ratios within the upper part of the basin range from 5 to 100 ("Methods"). Width-Depth ratio values ≥30 are within or adjacent to unchannelized segments, and W/D ratio values ≤30 are associated with channels (Fig. 2b). Furthermore, Width-Depth ratios calculated for Muskox Valley generally fall within the range of values reported for a wide range of alluvial rivers located in the Pacific northwest[33], and desert southwest[34] of North America. Significantly, however, Width-Depth ratios in Muskox Valley are unrelated to local valley slope, or to erosive power (Fig. 2b).

Although the mechanical controls (i.e., channel steepness and substrate erodibility) on channelization in Muskox Valley are inconsistent with conventional sediment transport analyses of erosion by surface water runoff, channel incision and architecture development are vigorous, nevertheless. Gravel bar-like features are, for example, present along Muskox Valley, indicating that peak seasonal streamflows over the past six decades or more were capable of transporting relatively coarse sediment particles (Supplementary Figs. 2 and 3). However, the valley profile has convex and concave segments, suggesting that erosional and depositional processes presently occur at specific locations along the valley floor, respectively, and that sediment transport potential has strong spatial signatures along the valley floor (Fig. 2b). These observations lead to a question: What processes govern the production and discharge of water with a sufficient flow rate to enable extensive channelization within Muskox Valley, and to what extent are these processes modulated by the seasonally varying thermal state of the substrate? This question directly addresses a critical knowledge gap related to sediment transport mechanics in permafrost and cold regions[18], which at present, has been little explored compared to more temperate landscapes.

We propose that the supply of water to emergent channel segments in Muskox Valley is governed by both point and spatially diffuse sources. The low elevation of the lake surface in 2019 relative to the channel head and the positions of beach terraces above the lake surface height suggest that intermittent floods from the lake (a point source) contribute to channel evolution, a topic we discuss below. Furthermore, our data suggests that post-glacial incision of Muskox Valley involves seasonal[4–10,19–21,35] water-delivery processes acting in concert to varying extents over the full length of the basin (distributed sources). For example, several prominent tributaries merge to the main valley at locations where unchannelized wetland segments occur (Fig. 2a). Tributaries act as seasonal surface and subsurface water reservoirs by storing and (re-)releasing water to downstream reaches throughout the summer season[6,9,19,21]. Over multiple seasons, presumably depending on the intensity of summertime warming as well as rain and/or snowfall events[4–6], there are evident effects of drainage from the fronts of earth flows on the south side of the valley (Fig. 2a and Supplementary Fig. 2) and solifluction lobes on both sides of the valley (see gullies and water tracks[10] within inset photograph Fig. 2a).

An aerial photograph from 1959 reveals important clues regarding the general pace, time-dependence and spatially patchy nature of periglacial landform development and channelization within Muskox Valley (Fig. 4 and Supplementary Fig. 3). In 1959, one channelized segment occurred downstream of the lake up and until the point of strong valley narrowing (Figs. 4a and 1, respectively).

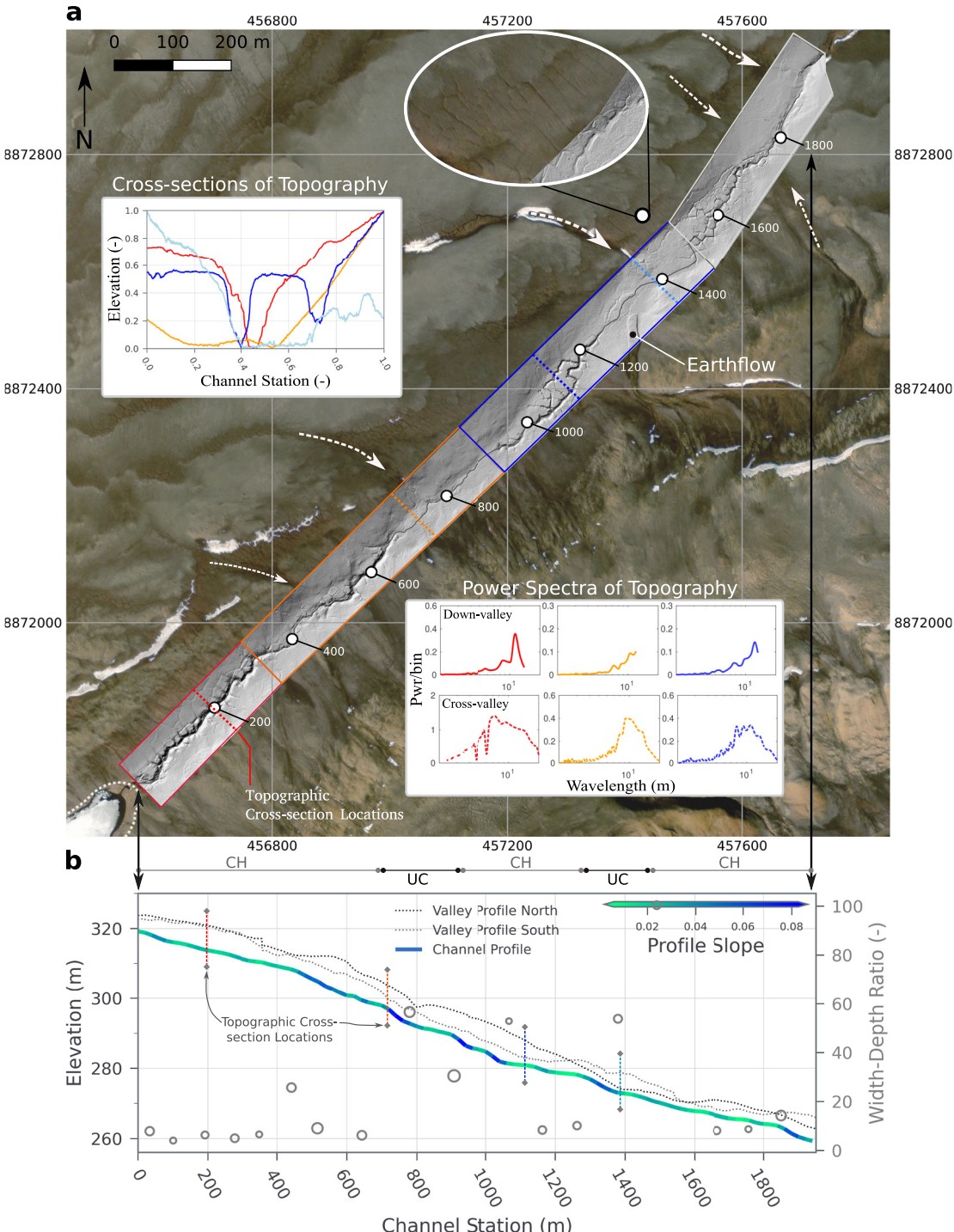

**Fig. 2 | Muskox Valley LiDAR and topographic profiles. a** LiDAR-based hillshade spanning 2000 m of Muskox Valley coincides with the dashed white outlined area in Fig. 1. Top inset figure shows topographic cross-sections oriented perpendicular to the direction of flow located at the color-keyed dashed lines along the hillshade. Cross-sections have been normalized to the maximum elevation and ground station. The bottom inset figure shows power spectra of down- and cross-valley topographic profiles ("Methods"). The spectra are color-keyed to the areas denoted by the solid colored boxes along the hillshade. The top inset photograph shows the nature of gullied hillslope drainage emerging from the fronts of solifluction lobes. The white-filled dot is located in the center of the inset image. Channel stationing at 200 m increments is indicated by white dots and corresponds to the channel and valley profiles shown in 2b. A thin black line overlain on the hillshade traces the channel profile. Dashed white arrows along the valley margins highlight locations where tributaries merge into the valley; arrow thickness proportional to tributary drainage area. The white dotted line segment around the lake indicates inferred lake surface elevation in summer 1959 (see Fig. 4). Satellite image sources: Esri | Maxar (DigitalGlobe) | Earthstar Geographics | GIS User Community, Imagery ID: 10df2279f9684e4a9f6a7f08febac2a9. Coordinates are UTM projection, Zone 16. **b** Channel and valley elevation profiles derived from the LiDAR data, and valley-bottom Width-Depth ratios (open circles) ("Methods"). Channel elevation profiles correspond to the stationing line shown in 2a; valley profiles are drawn parallel to the valley orientation outside of the channelized regions of the LiDAR. The colors of the channel elevation profile correspond to the local average slope, calculated over a 20 m moving window. CH stands for channelized, and UC stands for unchannelized.

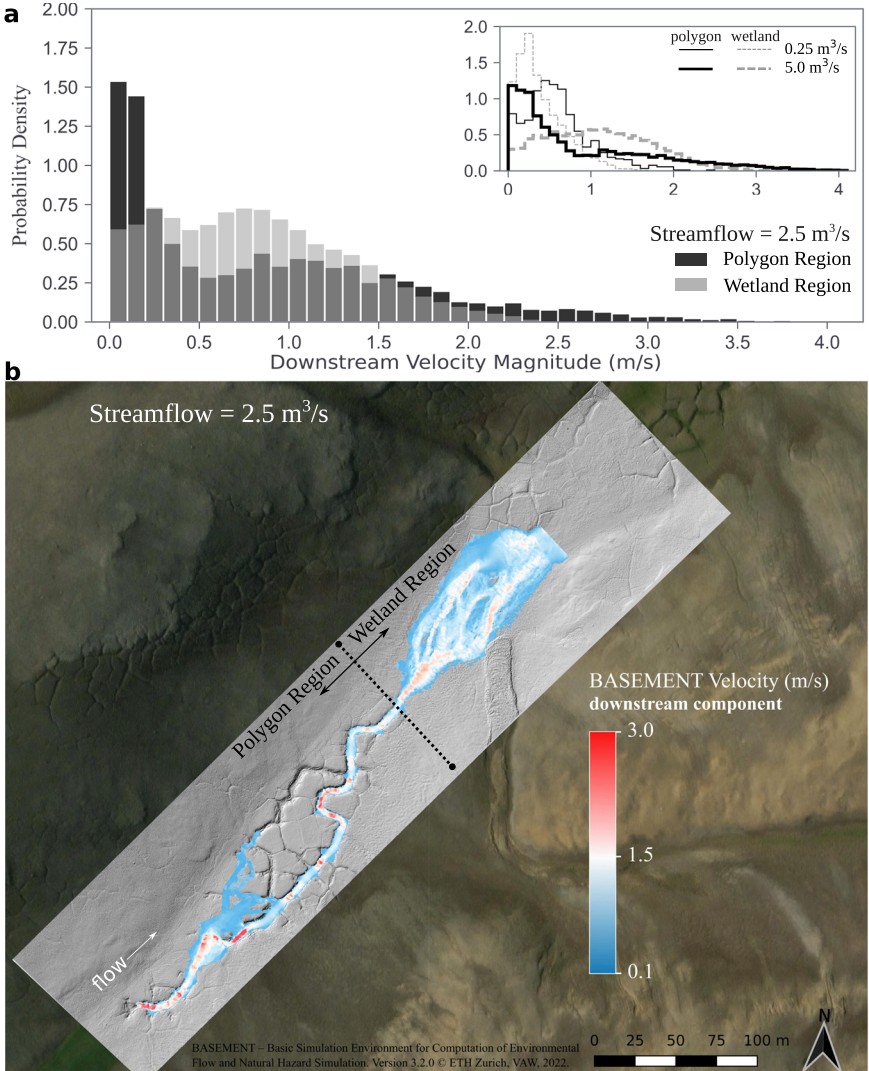

**Fig. 3 | Water runoff routing within Muskox Valley. a** Discrete probability distributions of the locally averaged downstream velocity for streamflows of 0.25, 2.5 and 5.0 m³/s along the valley floor from station 900 to 1400 m (Fig. 2a). Downstream velocities smaller than approximately 0.09 m/s are not shown in the distributions. Streamflows of 2.5 and 5 cm represent plausible estimates of lake outburst flood peak flow magnitudes depending on breach geometry ("Methods"). The distributions have been categorized based on whether a cell occurs within the polygon or wetland regions. **b** Spatial distribution of locally averaged downstream velocity for a streamflow of 2.5 cm along the valley floor from station 900 to 1400 m (Fig. 2a). Water runoff in the polygon region is spatially organized within interconnected polygon troughs and influenced by local topographic variation, whereas within the wetland region runoff spreads across the surface and is influenced by surface roughness. The simulations were completed without including bedload sediment transport, the combination of which is commonly referred to as morphodynamics, or when surface flows induce a topographic change of the channel bed surface and banks through net sediment particle erosion and deposition. The simulations were conducted using BASEMENT-Basic Simulation Environment, ETH Zurich[52] ("Methods"). Satellite image sources: Esri | Maxar (DigitalGlobe) | Earthstar Geographics | GIS User Community, Imagery ID: 10df2279f9684e4a9f6a7f08febac2a9.

The pre-existing channel segment had a rounded and entrenched geometry with near-constant channel width. These characteristics are compatible with existing studies of collapse features related to the melting of spatially discontinuous ground ice in periglacial soil profiles, as well as the melting of ice wedges along polygon troughs[7,8]. Areas upstream and downstream from this pre-existing channel segment in Muskox Valley were mostly unchannelized and polygons were generally absent (Supplementary Fig. 4). Over the next six decades, valley floor topography was profoundly altered through the development of multiple polygon fields, additional channelized segments, and activation of one earthflow (Figs. 2–4). For example, the 350 m long valley segment immediately downstream of the lake developed well-defined polygons and channelized during the intervening 60 years (Fig. 3b). This polygon control over channel incision and evolution is less evident in the 1959 image

data, and as a result, we hypothesize that coupling between polygons and enhanced valley channelization emerged with changes in the Muskox Valley hydrologic system related to the acceleration in Arctic warming evident since the 1980s[1–3,11] (Fig. 1). Such a sensitivity of polygon growth to, for example, enhanced groundwater flow is expected from physical models of frost heave applied to Arctic environments[25].

In addition to increased production and release of meltwater, a comparison of the 1959 and 2018 base imagery suggests that at least one lake outburst flood occurred[36,37] (Figs. 2a and 4b). We infer that the emergence of polygons alongside the downstream margin of the lake contributed to this flood due to lake leakage along contraction cracks, with ultimate failure and channel initiation in response to rising lake levels associated with climate change. Differences between the post-1959 and 2019 lake level positions suggest that approximately

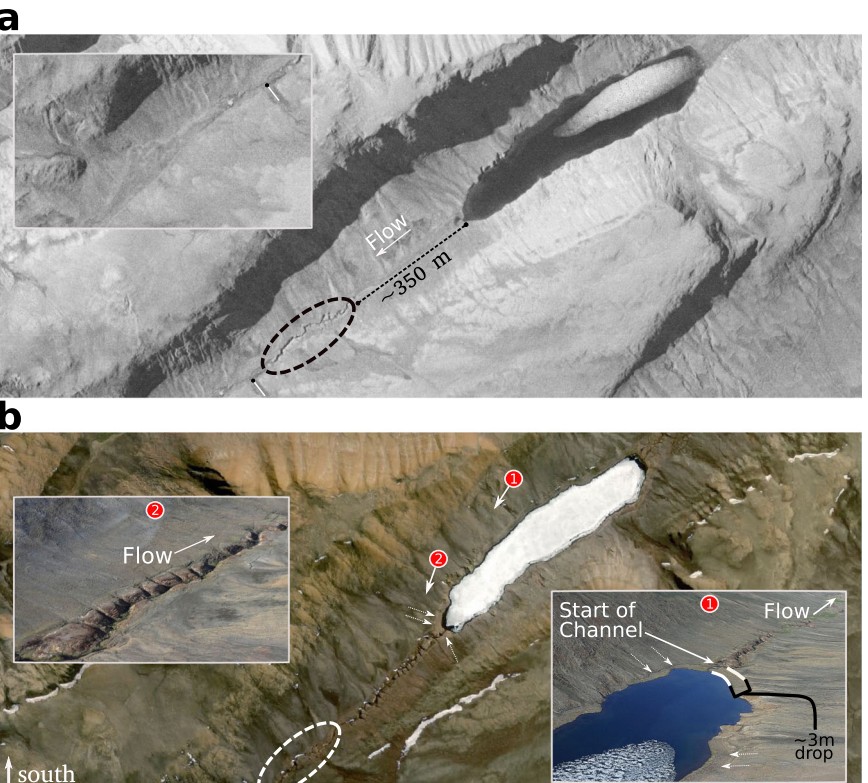

**Fig. 4 | Muskox Valley in 1959 and 2018. a** Muskox Valley in 1959 within the upper part of the LiDAR coverage of Fig. 2a. Circular dashed region indicates the extent of channelization near the lake at the time the image was taken, located approximately 350 m downstream of the lake. The inset photograph shows a zoomed-out part of the valley floor downstream of the channelized segment. A solid white line perpendicular to the flow direction with a black dot tip is shown for reference with respect to the inset image. The location of the inset image is shown in Supplementary Fig. 4. Image source from National Air Photo Library of Canada, roll and image #A16860_042; copyright holder: Natural Resources Canada. **b** Same location in Muskox Valley in 2018. Numbered circles with arrows indicate the location and camera orientation of respective inset photographs (taken by authors). The circular dashed region matches that in Fig. 4a. Dotted white arrows show a former lake shoreline that is approximately 3 m higher than the lake level during our visit in July 2019 based on the LiDAR coverage. Satellite image sources: Esri | Maxar (DigitalGlobe) | Earthstar Geographics | GIS User Community, Imagery ID: 10df2279f9684e4a9f6a7f08febac2a9.

~$10^4$–$10^5$ m$^3$ of water was released into the valley during the flood ("Methods"). Assuming a single outburst flood occurred, simple calculations on the basis of the approximate transport properties of sediment particles associated with plausible flow geometries suggest that the flood was capable of transporting gravel and cobble-sized sediment[31,32], resulting in gravel and cobble imbrication observed at various locations along the channelized reaches ("Methods"; Supplementary Figs. 2 and 3). As a result, the flood event likely enhanced channelization and deposition patterns downstream of the lake, yet did so without destroying or modifying the isotropic pattern of the polygons (Figs. 1–3).

### Inferred stream network development as a response to accelerating climate change

The combination of imagery and field observations supports the hypothesis that stream network development within Muskox Valley over six decades arises from a complex set of interconnected physical responses to amplified Arctic warming[1–11,19–23,27] (Fig. 5). The emergent polygon control on channelization may reflect the dynamics of ice wedge[11,24] and lens formation[25,26], each of which is sensitive to a surface climate, and surface/subsurface hydrologic system that is evolving with global warming[9,11,19] (Fig. 5b–d and 5g–i). Surface flow can emerge at channel heads and sidewalls, as well as at failure scarps where hydraulic forces can cause erosion and mechanical failure[16,17] (Supplementary Movie 1), and channel migration and widening (Fig. 5c, d and g), supplying sediment to down gradient locations. The particle size distribution of these sediments is diagnostic of the textural properties and erodibility of their sources and the intensity of the water flow[18]. Compared to sediments sourced from retrogressive thaw slumps, solifluction lobe fronts, and hillslopes, relatively more vigorous surface flows confined among polygons (Fig. 3), and related to lake outburst floods or glacial melting events are more likely to entrain and transport coarse sediments where available (Fig. 5b–d, g, i, j), depending on the extent to which the bed over which they flow is frozen (Fig. 5d inset and 6).

Increases in average summertime air temperatures over at least the last four decades increase wet precipitation[6,38] and storage in lakes (Fig. 5j), as well as the seasonal production of soil melt water[6,19,20,35]. Accelerating global warming also increases the thickness of active layer soils involved in seasonal freeze-thaw cycles[6], and the vertical extent of ground ice melting[6,19,20,35] (Fig. 5b, 5d–j). These responses influence the decadal-scale delivery of subsurface meltwater and consequently sediment into tributaries and channels[21], as well as the extent of water draining from solifluction lobes (Fig. 5i). The loss of pore ice reduces soil strength and also enhances soil permeability[19], which augments water transport at the expense of storage[6]. These effects enable a greater seasonal delivery of water to the surface with Arctic warming[6,9,20,22,38], which may be further enhanced as a result of proportionally more rainfall in annual precipitation totals[38] (Fig. 5).

Our study highlights that landforms such as polygons influence the path of surface water flow in Muskox Valley (Figs. 1–3), consistent with observations of channel systems on Bylot Island, NU[7,8], the Pan-Arctic in general[11], the Antarctic Dry Valleys[22,23] and Mars[30]. More broadly, evolving configurations of polygons, retrogressive thaw

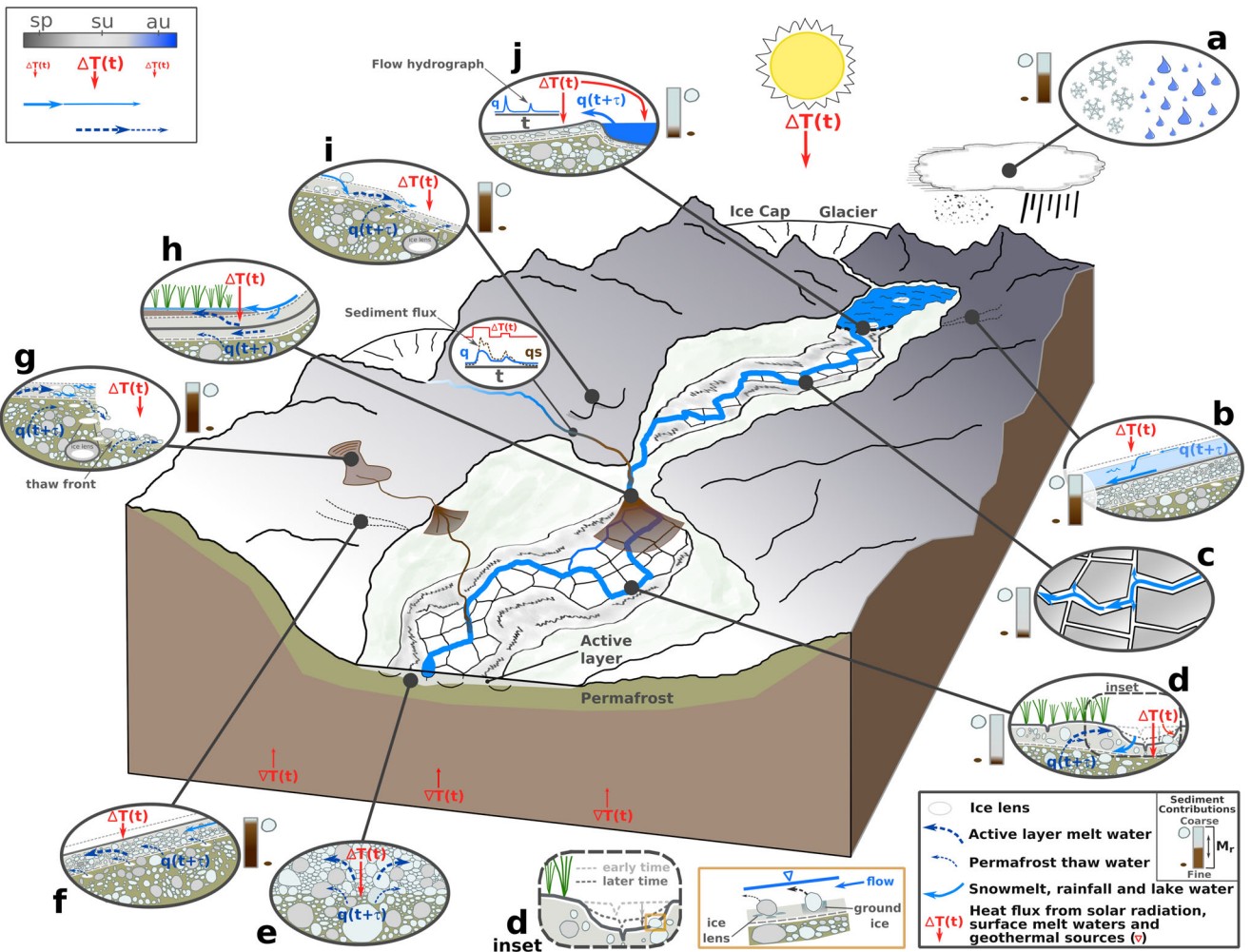

**Fig. 5 | Valley and local-scale processes that contribute to channelization in permafrost settings under a warming climate. a** Precipitation in Muskox Valley is partitioned between snow and rain. The balance between these two phases may shift under a warming climate[6,20,37]. The seasonal energy budget of the ground surface is a function of the incoming solar radiation. Geothermal flux contributes energy to the base of the permafrost. **b** Snowmelt runs over the snow surface for relatively short distances and then vertically into the snowpack, where it can move down slope following multiple possible flow paths[39]. The dominant flow path is assumed to be at the base of the snow on the ground surface, in contact with the uppermost active layer substrate. The flow path taken influences the arrival time of snowmelt components to the basin[39]. **c** Down-valley channel pathways develop randomly as water flows down gradient through polygon troughs. The occurrence of polygons has a strong control on channel path selection[7,8,11,22,23,30]. **d** Water flows into polygon troughs and thermal contraction cracks and down the channel network, eroding the boundary and delivering heat to the active layer and the top of the permafrost. Channel boundary erosion and seasonal thermal changes to the active layer accentuate channel development. Heat delivery to the top of the permafrost and the active layer produces thaw and melt water which can move back into the channel network and add to water runoff. d inset. Channel incision and cross-section development occur over time influenced by the timing and occurrence of sediment transporting flows, and relative mobility and erosivity of sediment particles resting on the river bed surface[18]. Sediment particle mobility and erosivity in permafrost setting rivers are uncertain due to a lack of understanding of how thermal conditions of the particle bed influence erosion mechanics (Fig. 6). The relative abundance of ice lens and ground ice are two possible controls, including the coincidental timing of potentially erosive flows. **e** Sediments in the valley floor deposited under previous glacial and interglacial periods. Sediments in the permafrost are ice-matrix supported. Seasonal permafrost thaw contributes to local water runoff and channel development[6,22]. **f** Gravity-driven flows within the active layer of swales occur on valley slopes, delivering water to the valley floor. Water delivery lags the delivery of heat from the surface. Swales may contribute to channel network development by providing paths for tributary growth. **g** Retrogressive thaw slumps occur on valley slopes. Thaw slumps are sources of seasonal water and sediment to the downstream basin[3–6] and, like swales, may contribute to channel network development by providing paths for tributary growth[6]. **h** Coarse and finer sediment particles are deposited to form wetlands and fans where flow emerges from valley constrictions and possibly in association with tributary junctions. Wetland development leads to an increased number of shallow groundwater flow paths and locally more complex surface-groundwater interactions. Polygonal terrain can be buried in the process. **i** Solifluction lobes are common along permafrost valley wall slopes. Their complex stacked geometry can disrupt downslope water movement locally, and their water discharge at their snouts provides water seasonally to the downstream basin. **j** Valley bottom lakes intermittently contribute flood pulses to downstream channel networks. Flood pulse flow rate per unit area is large relative to other water sources and contributes to channelization through sediment transport and boundary erosion. The legends provide context for the fluxes shown in the vignettes, with sp: spring, su: summer, au: autumn. The abbreviation $M_r$ stands for the relative magnitude of sediment supply of fine (suspended load) and coarse (bedload) sediment size fractions. The relative magnitude is assigned to indicated vignettes based on the literature[6–10,18–23,31,32], observations reported here, and the generally reported result that the suspended load of a river is larger than the bedload flux. The figure was inspired by schematics developed for gravel-bed floodplain environments[53] and complements other conceptual frameworks for cold region geomorphology[9,18,19].

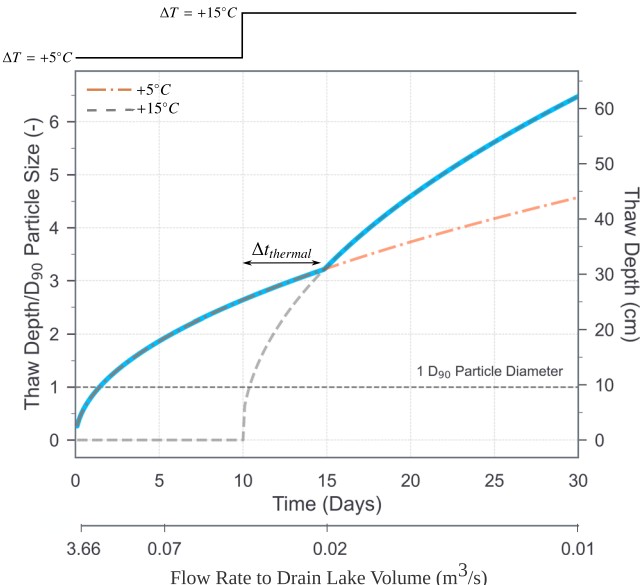

**Fig. 6 | Active layer thaw depth estimate.** Thaw depth scaled by the $D_{90}$ Particle Size = 0.096 m for two surface air temperature warming events. The blue curve indicates a thought experiment where a typical July surface air temperature increase of 5 °C drives active layer thaw conductively for 10 days before a relatively sharp (stepwise) 10 °C warming event is imparted at the surface. The thermal effect of this powerful warming event diffuses to the depth of the melting front over about 4 days, driving enhanced penetration of the melting front over the rest of the month. This step increase in surface temperature increases the depth of the July thaw by approximately 40%. In addition to increasing the depth of the active layer thickness by the end of the melt season, which enhances potential soil mobility, an important consequence for the subsequent melt seasons is an increase in the soil porosity and, in turn, the potential for enhanced subsurface hydrologic storage[19] and potentially steeper hydraulic gradients in and around channelizing reaches within interconnected polygon trough regions. We propose that enhanced melting with channelization could introduce a positive feedback to this overall process, which would effectively contribute to advancing the seasonal thaw front even deeper into the ground, driving additional permafrost degradation[11]. In the Muskox Valley context, we also provide an estimate of the flows associated with the equivalent lake draining time scale, assuming a lake volume loss of $10^4$–$10^5$ m³ ("Methods"). Calculations assume the thermal properties of a saturated, vertically homogeneous gravelly-sand substrate. See the "Methods" section for the calculation procedure using a solution to the Stefan problem[49] scaled for the volumetric water content, and the online repository for a Jupyter Notebook detailing the parameter values, calculations and plotting of this figure.

slumps and solifluction lobes, along with active layer thicknesses and structure, influence the availability and delivery of water to the channel network by changing surface and subsurface flow paths[6,9,11,19] (Supplementary Movies 1 and 2). In turn, the production, erosion, transport and deposition of fine and coarse sediment (Fig. 5b–d, f, g, i and j) stemming from these climate-controlled hydrologic processes drive topographic adjustment of valley walls and margins, as well as the valley floor and channel surfaces[18,27].

These landscape evolution processes are sensitive to peak flow magnitude[31,32], when in the thaw season peak flows occur, and the total duration of seasonal and event-based surface flows. The extent to which such hydrologic events drive erosion and landscape change is controlled by the relative mobility and erosivity of sediment particles resting on, and near the surface, which are, in part, functions of the particle-bed thermal state (Fig. 5d inset and 6). In particular, the penetration depth of melting fronts related to surface water flows is proportional to the square root of the magnitude of the surface temperature forcing, and the time scale over which events occur ("Methods"). Thus, the extent to which erosion is thermally controlled depends on the intensity of seasonal surface temperature variability,

and event duration. For example, impulsive thermal effects of hour-to-day-long floods penetrate only ~1–10 cm into the channel bed substrate, roughly the diameter of the larger size cobbles on the bed surface (Fig. 6). Thus, whether such floods erode into the bed surface and transport coarse sediments downstream depends on event timing during a particular melt season. If a flood occurs after a ~10-day period of snowmelt, the mobility of particles on the bed surface and the depth of erosion into the bed is enhanced (Fig. 6). By contrast, if flooding occurs before significant snowmelt, erosion will be inhibited by a relatively much stronger bed. Future efforts to better understand channelization in permafrost settings should focus on disentangling the erosional mechanics of frozen, or partially frozen substrate[18], including complex but probably important particle-scale processes such as saltating sediment particles, mechanical and thermal effects of which can likely enhance sediment mobility at the surface, and hence channel bed erosion (Fig. 5d inset).

Last, locations where silts and clays are deposited lead to seasonal wetlands with confined and semi-confined shallow flow paths (Fig. 5h). These wetland locations act as local water sinks and modulate the pace of valley profile change by destroying erosional signals through local deposition. Interactions and cascading effects among the various processes illustrated in Fig. 5 occur over time scales on the order of years to decades[6,9,19,21] and can have strong local variations at length scales on the order of the valley floor width, bounding hillslope and individual polygon (Figs. 1, 2 and 4).

## Implications of our observations and looking ahead

The dramatic and apparently recent channelization response of Muskox Valley to accelerating global warming is probably not unique[9,11,18,19,21]. We documented similar landscape attributes at numerous locations on the eastern side of Axel Heiberg Island, and near the Haughton impact structure of Devon Island (Supplementary Fig. 5; Supplementary Table 2; Supplementary Movie 2). Additionally, multi-decade hydrometric data from the Tibetan Plateau suggests that rapidly warming regional air temperatures are responsible for an increase in measured water and fine sediment flux through a control on the seasonal active contributing drainage area[21]. Consequently, Fig. 5 makes specific predictions that are broadly applicable to arid, permafrost-rich Arctic and high-altitude environments. Increased seasonal and annual surface water runoff accentuates channel erosion and, as a result, the spatial expansion of stream networks[7,8]. The seasonal timing of these processes is strongly influenced by complex interactions among hydrologic sources[18,27] and the length and complexity of travel pathways in periglacial environments[6,9,19,21,39]. For example, permafrost thaw and pore ice melt waters that travel down hydraulic gradients, through the active layer and discharge as seepage flows at retrogressive thaw slump scarps over time scales of hours to days (Supplementary Movie 1), or at polygon trough walls. Polygons have an emergent basic control on where components of channel networks form and evolve by spatially organizing seasonal water sources[7,8,11,22,23] (Fig. 3 and Supplementary Fig. 1), which interact and coalesce down valley into nascent and evolving channel segments, the locations of which are set by trough positions and orientations (Figs. 1 and 2). Our work shows that periglacial landform control on channelization can remain resilient to hydrologic events across multi-decadal to century time scales, which implies periglacial landforms are fundamental to processes of channel network establishment in permafrost-rich locations undergoing relatively rapid changes related to a warming climate[11].

Moving forward, a key challenge is to use observational data akin to that presented here to develop predictive physical models of the time-varying interconnections among the soil pore (cm) to valley (km) scale landscape change processes we infer and characterize at Muskox Valley, and more generally within relatively arid, permafrost-rich Arctic and high-elevation environments[6,9–11,18–21].

Testing hypotheses of these interconnections (Figs. 3, 5 and 6) will build an understanding of the spatial and temporal response properties of these environments to effects of accelerating climate change[11,38,40] (Fig. 1 inset), and amplified summertime warming variability[41,42]. On the basis of our results, an important feature to explain is the coupling among climate, polygons, channelization and permafrost degradation (Figs. 3–5). Under the present warming climate, our work suggests that polygons locally enhance rates of channelization through surface flow organization (Fig. 3; Supplementary Movie 2), which increases mechanical erosion of valley floor substrate, thermal degradation of ice wedges and subsequent ground collapse over decadal time scales[7,8,11] (Fig. 5d). Over this time, these combined processes can drive channel incision to depths of several meters, or more. Channel cross-section development within polygon troughs locally steepens hydraulic gradients, and a deepening channel provides increased surface areas at the channel margins for heat exchange, which can, in turn, increase rates of local permafrost thaw. These cascading effects can enhance the release of greenhouse gases within these environments as organic soil carbon thaws in response to permafrost retreat[28] (Fig. 5d). A similar effect can occur through hillslope gully formation related to persistent seepage from solifluction lobes (Fig. 2a), and from the erosive action of retrogressive thaw slumps (Supplementary Movie 1), causing the downstream delivery of labile particulate organic carbon[6,18]. By embracing a full bandwidth of underlying periglacial processes that govern the character and cadence of environmental forcings related to Arctic climate change and its variability, our work offers a local framework to develop new generations of physical models of valley-scale channel network development, helping to raise recognition and awareness of process-based positive feedbacks with climate warming[6,9–11,18,19,27,28,40].

## Methods

### Additional field site description

Muskox Valley (8872800, 457600, UTM projection, Zone 16) is located on Axel Heiberg Island, part of the Qikiqtani Region, NU, the Canadian Arctic Archipelago (Fig. 1). The field site occurs within a recently deglaciated valley on the eastern side of the island, and lies a few kilometers to the southeast of a prominent ice lobe extending from the Müller Ice Cap. Regional deglaciation began between 8 and 10.3 kyr ago[29].

Muskox Valley is 8 km long, and 1 km wide, with a flattened U-shape. The valley floor ranges in elevation from 140 to 320 m asl and consists of steeper and flatter segments, presently characterized by developing channels and wetlands (Figs. 1 and 4 and Supplementary Figs. 2 and 3). Field inspection of exposed deposits within the channel banks and substrate at the ground surface suggests the valley is underlain by glaciofluvial sediments ranging in diameter from sand- to cobble-sized particles (Supplementary Figs. 2 and 3). One lake is located within the upper part of the basin, with channel development focused downstream of the lake to the basin outlet (Fig. 1). These combined physical characteristics of Muskox Valley suggest that the valley was a subglacial channel, draining meltwater from a localized part of the Müller Ice Cap during the last glacial period[29].

Eastern Axel Heiberg Island has a polar desert climate, which includes ice caps and non-ice-covered regions. Climate data collected at Eureka, NU, Canada for the period 1981–2010 indicates 50–100 mm per year of annual average precipitation, and an annual average air temperature of −19 °C (Environment Canada, World Meteorological Station ID 71917). Precipitation is split between about 60% snow water equivalent and 40% rainfall. Air temperatures during fieldwork in July 2019 were between 15 and 20 °C, compared to a July average of 6 °C for the period 1981–2010 (Environment Canada, World Meteorological Station ID 71917).

### Capturing and processing of LiDAR point clouds

We collected surface topography data in Muskox Valley during a one-day field campaign on July 7, 2019, using the AkhkaR4DW backpack mobile laser scanning system[43,44]. The portable unit collects high-precision 3D topographic data kinematically[44] at a maximum spatial resolution of 0.1–1 cm. The system operation is based on Global Navigation Satellite System–Inertial Measurement Unit (GNSS-IMU) positioning and observes GPS and GLONASS constellations for position. Precise platform movements are observed by the near-navigation grade IMU to produce attitude data for precise georeferencing of the laser scanner and image data. 3D data collection is carried out by two profiling laser scanners, each operating at different wavelengths[44]. The trajectory data was post-processed with data from a base station located at our camp roughly 60 km south of Muskox Valley at coordinates 8813520, 482220, UTM projection, Zone 16.

We mapped roughly 2 km of the valley floor extending downstream from the relict glacial lake, with a valley floor width of approximately 200 m. Mapping occurred by walking across the valley floor, roughly following the polygon contour and other terrain features to collect seamless data over the valley. Within channelized reaches, we followed the channel centerline and then mapped each bank by climbing in and out of the channel along collection lines with a spacing of about 5 m. From median elevations binned and resampled onto a 2 cm grid, we use LASTOOLS (LAStools, Efficient LiDAR Processing Software [version 191018, academic], obtained from http://rapidlasso.com/LAStools) to interpolate the data to produce DEMs with a 4 cm resolution.

### Power spectral estimation

We first extract ~20–100 down-valley and cross-valley profiles from each of the LiDAR image tiles that are combined to produce the Hillshade in Fig. 2. A 1D power spectrum is estimated to a 95% confidence for each profile using a Thompson multitaper method[45]. As additional checks, spectral results for the fundamental mode discussed in this paper are then compared with estimates from conventional Welch and autocorrelation-based methods. Average down-valley and cross-valley spectra are then binned and smoothed to produce the results in Fig. 2. Down-valley and cross-valley polygon field isotropy is at the well-resolved largest scale of the ~10 m fundamental mode. Power spectral analysis of the ground topography is useful because it characterizes the dominant scale(s) of the topographic ground structure(s), or physical patterns. In the present case, the power spectra reveal that polygon fields along the valley floor have a similar dominant length scale measured both down and across the valley. Furthermore, and depending on how polygons are spatially organized, this dominant length scale imparts a relative and consistent unit flow path length as surface flows incise apparently stochastically along interconnected polygon troughs.

### Calculation of width-depth ratio and local channel gradient

We acquired valley cross-section profiles on LiDAR elevation data to calculate width-depth ratios. We used ArcGIS 10.8.1 to clip the LiDAR data in four 600 m wide tiles along valley, measuring 1–2 km each. We then extracted 20 cross-sections spaced regularly every 70 m down valley, which were then loaded onto MATLAB to calculate cross-section slope and curvature (Supplementary Fig. 6). To minimize uncertainty related to qualitative measurements, we measured top width as the distance between curvature minima, and bottom width (valley floor) as the distance between curvature minima located within the previously identified valley width[46]. Note that some profiles have multiple channels. These were also measured following the same technique. Associated depth along each transect was measured as the deepest point from the top width line.

Channel gradient calculated from the processed LiDAR point cloud as a centered average for a window length of 20 m. Averaging

window does not overlap, and represents the gradient for discrete 20 m channel segments, beginning at the upstream extent of valley channelization.

### Calculation of the outburst flood volume, flow rate and mobility analysis

The ESRI 30 cm Hi Resolution World Imagery (ESRI Map Service, ID: 10df2279f9684e4a9f6a7f08febac2a9) and the 2019 LiDAR data were used to estimate the volume of water associated with a proposed lake outburst flood event that occurred sometime between 1959 and 2019. The pre-event lake surface area measured approximately 95,000 $m^2$: 750 m × 150 m × 0.85, where 0.85 is a shape factor correction from a rectangle. The post-event lake surface area measured approximately 60,000 $m^2$: 700 m × 100 m × 0.85. The differential area associated with the lake outburst flood is approximately 35,000 $m^2$. Our LiDAR and photographic data suggest between 1959 and 2019, the lake water surface elevation dropped by approximately 3 m (Fig. 4b). If this entire depth loss is associated with the lake outburst flood, the estimated flood volume is within the range of $10^4$–$10^5$ $m^3$. It is also possible that all or a proportion of this volume was lost due to slow leakage at the downstream lake margin. Here we focus on the lake outburst loss mechanism as it provides a plausible explanation for the balance of our field and photographic observations, and inferences—in particular, the imbrication of gravels and cobbles along channelized segments.

We used an empirical relationship[47] to estimate the outburst flood peak flow rate as $Q_p \sim 3.1 W^*H^{1.5*}[A/(A + \tau^*H^{0.5})]^3$, where $W = W_b + Z_b^*H$ with $W_b$ the breach width, $Z_b$ a parameter with a value of 1.0, $H$ the height of water above the breach, $\tau$ the elapsed time since breach development, $A = 23.4 S_a/W$ with $S_a$ the surface area of the lake at the breach elevation (Supplementary Table 3). For the calculations, we assume an average lake surface area of 72,500 $m^2$ (average of 95,000 and 60,000 $m^2$), a breach width of 2 m, a breach hydraulic head of 1 m and a breach development time of 0.167 h. Calculations are done using imperial units and converted to metric.

Field photographs suggest that the peak flood depths of the proposed lake outburst flood could have achieved 2 m; we also recognize that this is difficult to discern in the field as incision likely occurred during the passage of the flood peak and thus depths could be overestimated from field evidence. Despite the uncertainty associated with the thermal state of bed sediments when the proposed flood may have occurred, we use a simple Shield's analysis[31,32] to estimate conditions necessary to mobilize gravel and cobble size sediment particles observed on the streambed in Muskox Valley for flow depths of 1 and 2 m (Supplementary Figs. 2 and 3). The visual and photographic estimated 50th and 90th percentile grain sizes on the streambed in the valley were 20 and 96 mm. The slope-corrected[48] critical Shield's condition using an average valley slope of 0.03 suggests flow depths of 2 m are sufficient to transport both the 50th and 90th percentile grain sizes, and flow depths of 1 m are sufficient to transport the 50th percentile size (Supplementary Table 1). We assumed top flow widths of 2 and 3 m in both cases and applied a cross-section shape adjustment factor of 0.5 to calculate the hydraulic radius.

### Calculation of estimated active layer thaw depth

Active layer thaw depth estimate made using a solution to the Stefan problem[49] scaled for the volumetric water content: $L\_skin = \lambda^*[(2k\Delta T_s t)/(\theta \rho_w L)]^{0.5}$ where $\lambda$ is a correction factor[49] with a value less than 1, $k$ is the thermal conductivity of the bulk unfrozen substrate which is assumed to be a saturated, vertically homogeneous gravelly sand, $\Delta T_s$ is an air temperature step change at the surface above 0 °C, $t$ is the time since the start of air temperature

step change, $\theta$ is the volume fraction of water in the substrate, $\rho_w$ is the bulk density of water and $L$ is the mass-based latent heat for the fusion of water. Calculations based on $k = 1.1$ W $m^{-1}$ $C^{-1}$ per Dalla Santa et al.[50], $\theta = 0.40$, $\rho_w = 1000$ kg/$m^3$, $L = 3.34^*10^5$ J/kg, and $\lambda$ has a value of 0.95 based on Fig. 5 of Kurylyk and Hayashi[49]. In choosing the thermal conductivity of the gravelly sand substrate, we assume no appreciable shallow groundwater flow. In the context of Muskox Valley and similar landscapes, the time scale t is that for a lake draining event $V_f/Q_a$, where $V_f$ is the estimated volume of water released during the outburst flood event (discussed above), and $Q_a$ is the average flow rate ($m^3$/s) during the flood event. The time scale $V_f/Q_a$ gives the time over which a thermal perturbation is delivered to the active layer substrate. See the online repository for a Jupyter Notebook documenting the calculations and plotting of Fig. 6[51].

### Hydrodynamic modeling

The simulations were conducted using the LiDAR-derived DEM, and BASEMENT-Basic Simulation Environment, ETH Zurich[52]. We focused the modeling on a portion of Muskox Valley which is channelized and also supports a wetland. This choice was made in order to examine how differing landform properties influence the routing of surface flows. The input unstructured mesh was calculated using a plugin for QGIS for BASEMENT, with cell sizes ranging from 0.10–5 m. The mesh was delineated into two regions of channel and floodplain, or overbank. The hydrodynamic module solves the shallow water equations, where the system unknowns are the water surface elevation and the two components of the streamflow per unit width[52]. The hydraulic friction closure was calculated using the Manning-Strickler relation, with standard roughnesses set for the channel and the floodplain. Eddy viscosity was calculated with the constant model. The simulations assume no sediment transport and hence a fixed topographic boundary because we presently lack theory to estimate sediment particle entrainment of partially or completely frozen channel bed surfaces. Three different steady flows were assigned at the upstream boundary of 0.25, 2.5 and 5 $m^3$/s. The input mesh and model build files have been provided in the supporting online repository[51].

## Data availability

The processed topography data and the hydrodynamic model setup input mesh file have been deposited at figshare: https://doi.org/10.6084/m9.figshare.19790491.v1[51]. The repository includes a README.txt file to navigate the contents.

## Code availability

The Python scripts used to produce the inset to Fig. 1 and the main part of Fig. 6, and the hydrodynamic model setup files used to produce Fig. 3 and Supplementary Fig. 1 with the BASEMENT open access software are also provided in the above-referenced data repository at figshare. The calculations for Supplementary Tables 2 and 3 are provided as Supplementary Data 1 in the online supplemental information.

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

## Acknowledgements

This research was supported by a Canadian Space Agency Flights and Fieldwork for the Advancement of Science and Technology (FAST) Grant 18FAWESB26 (G.R.O. and A.M.J.); Natural Sciences and Engineering Research Council Discovery Grants (G.R.O. and A.M.J.), Natural Sciences and Engineering Research Council Discovery Accelerator Grant (A.M.J.) and Northern Research Supplement (G.R.O.); Natural Sciences and Engineering Research Council Postdoctoral Fellowship (S.M.C.); Arizona State University Exploration Fellowship (A.G.G.); NASA Postdoctoral Program at the Jet Propulsion Laboratory, California Institute of Technology, administered by Oak Ridge Associated Universities under a contract with NASA 80HQTR21CA005 (S.H.); Academy of Finland (Coe-LaSR, MS-PLS) and Strategic Research Council at the Academy of Finland projects 293389 and 314312 (A.K.). We thank the Polar Continental Shelf Program and their Resolute Bay staff for logistical support of our 2019 expedition. We thank the Inuit of the Qikiqtani Region (one of the three regions in the territory of Nunavut) for their permission to carry out fieldwork on their land in Nunavut and the community of Qausuittuq (Resolute Bay) for welcoming us during our stay in the summer of 2019. We encourage you to learn more about the land, communities and people of the Qikiqtani Region through the Qikiqtani Inuit Association (https://www.qia.ca/about-qikiqtani/).

## Author contributions

S.M.C. and A.M.J. conceived the study. S.M.C., A.M.J. and A.K. collected the high-resolution LiDAR data as well as the low-altitude and ground-based photographs in Muskox Valley. A.G.G. and S.M.C. identified valley systems with similar characteristics to Muskox Valley. A.K. processed the LiDAR data, and A.M.J. developed the digital elevation model from the point cloud. S.M.C., A.M.J. and A.G.G. conducted the data analysis, and S.M.C., A.M.J., A.K., A.G.G., G.R.O. and S.H. participated in interpreting the data. S.M.C. and A.M.J. wrote the manuscript with helpful comments from A.K., A.G.G., G.R.O. and S.H.

## Competing interests

The authors declare no competing interests.
