## [Peer Review File · Nature Communications]

High Arctic channel incision modulated by climate change and the emergence of polygonal groundREVIEWER COMMENTS

Reviewer #1 (Remarks to the Author):

I have reviewed the paper entitled 'Interplay among climate, polygonal terrain and channel formation in the Canadian High Arctic'. This study tries to reveal the interactions among climate change, hydrologic processes governing water delivery, and consequently the timing and pace of channelization and landscape change in a high Arctic permafrost valley setting. Personally, I think this is an important contribution to the existing literature as there is a critical knowledge gap in terms of measuring the sedimentary and geomorphic responses to modern climate change (East et al., 2022, *Earth's Future*), as well as the possible impacts of rapid erosion on carbon release and thus climate feedbacks (Zhang et al., 2022, *Nat. Rev. Earth Environ.*). The findings of this study would be of interest to a range of research communities including climatologists, geomorphologists, as well as hydrologists and permafrost scientists. Although some of the statements in this study seem speculative and lack more robust in-situ observations (details below), it will likely inspire and stimulate more follow-up studies to enhance observations in the Arctic and develop new generations of more rigorous physical models of channel network development in the strategically important cold regions.

Overall, I think the manuscript is likely suitable for publishing in *Nature Communications*, after clarification of the major questions and several aspects listed below.

(1) The study seems lack quantitative findings and some statements are vague. For example, in Lines 24-26: this sentence seems the key message of this study, but the readers are left unclear about the specific findings. e.g., what are the specific major controls over channel incision in permafrost landscapes? what are the plausible positive feedbacks among the climate, periglacial landforms and channelization processes? Lines 26-29: again, it is not clear here and seems overstated. What is the new generic landscape change model (it is still unclear after reading throughout the paper)? what does the 'intensifying variability' represent? Do you mean the 'intensifying variability of climate, runoff, or sediment supply'?

(2) Introduction (Lines 35-55): it does summarize the existing knowledge gaps in a concise and comprehensive way and does highlight the importance of the seasonal dynamics of climate, ice, the active layer, different water sources and sediment supply in the permafrost environment in driving landscape change and channel initiation, but this study lacks quantitative seasonal field observations of these key factors to support the underlying process. In particular, the seasonal water dynamics (multiple water sources in cold regions) and maybe also sediment supply can exert important controls on channel initiation and development. Maybe the data from similar permafrost landscapes would be helpful to explain some of the statements.

(3) The study concludes that it "offers a local framework to develop new generations of rigorous physical models of valley-scale channel network development....." However, the readers are left unclear about the framework and also the process-based positive feedback with climate warming. Could you please explicitly explain process-based positive feedback by inserting a conceptual framework in Figure 4? The current Figure 4 is nice and helpful, and it can be improved by highlighting the interactions of different processes.

(4) Similar to the High Mountain Asia (Li et al., 2021, *Science*), the Arctic climate has become warmer and wetter since the 1990s, with more summer rainfall that will likely change the hydrological processes and cause more permafrost disturbances and related erosion and channel network evolution. The changing rainfall patterns and their impacts on landscape changes (e.g., permafrost thermokarst development and channelization) should also be explicitly discussed (after a quick search, the word 'rainfall' only appears once in the main paper).

Additional comments:

(1) Line 79-81. This statement seems not supported by the data. Can you prove that the polygon geometry can influence or govern the channel path or paths both across and down topographic gradients at the valley scale?

(2) Line 137-139. Is it possible to provide more quantitative statements, such as the relative

contributions of different water sources to emergent channel segments? How frequent is the outburst floods from the upstream lake?

(3) Line 287-290. It is not entirely clear how steeper hydraulic gradients provide increased surface areas for heat exchange.

(4) Movie: unfortunately, the youtube link does NOT work and I did not see it from the manuscript tracking system.

(5) Method-Calculation of Width-Depth Ratio. It is not easy to follow here. Can you provide a sketch diagram for the calculation of width and depth? For example, how do you determine the average depth and width for the cross-sections in Figure 2?

(6) Method-Calculation of the Mobility Analysis and Table ED2. I am not exactly working on incipient sediment motion. Here are a few questions for your consideration. Is the Shield's analysis suitable to estimate sediment particle mobilization conditions here? how about other methods that are more suitable for pebble-cobble sized sediment particle mobilization? For Table ED2, the authors seem to treat the riverbed sediment as a uniform bed when calculating the grains' mobilization velocity for a certain D50 or D90. However, the actual channel is a non-uniform bed, and how to consider it? The study selects 4 sets of experimental calculations (Table ED2). Are they representative and can they truly reflect the critical flow conditions for riverbed surface particles? What about the different ice conditions that could affect sediment mobility?

(7) Figs. 1-3. Please add the scale bars in these images. Would it be good to highlight the geographical information on the figure instead of describing it in the captions (Fig. 1 and Fig. 3).

(8) Fig. 2. Could the fonts in this figure be a bit larger?

(9) Line 33: change 'mountain regions' to 'cold regions'?

(10) Fig. 4: nice figure. Maybe the long caption could be shortened a bit. I did not see the framework from this figure.

References mentioned:

East, A. E., Warrick, J. A., Li, D., Sankey, J. B., Redsteer, M. H., Gibbs, A. E.,... Barnard, P. L. (2022). Measuring and Attributing Sedimentary and Geomorphic Responses to Modern Climate Change: Challenges and Opportunities. *Earth's Future*, 10(10), e2022E-e2983E.
Li, D., Lu, X., Overeem, I., Walling, D. E., Syvitski, J., Kettner, A. J.,... Zhang, T. (2021). Exceptional increases in fluvial sediment fluxes in a warmer and wetter High Mountain Asia. *Science*, 374(6567), 599-603.

Reviewer #2 (Remarks to the Author):

Joel Rowland
Los Alamos National Laboratory

The manuscript provides a detailed field study of an evolving channel system in the Canadian High Arctic. Based on these observations that manuscript puts forward conceptual model for the relative controls of fluvial and thermal controls on erosion and channel development in permafrost settings characterized by ice wedge polygons. The paper is well written and the graphics are excellent. I think the study is of broad value and highlights future data needs in the field I reviewed an earlier submission of this manuscript and believe the paper is acceptable for publication in its present form. I only have a few minor points.

Figure 2: It is stated that the cross-section elevations are normalized by the maximum elevation. The axis reports elevation in m, but ranges from 0-1, are all of the cross sections only 1 m in height or should they be unitless? As mentioned in the prior review, it would be helpful if the manuscript explained what is learned by the power spectra and how it relates to the valley and channel evolution.

Line 165-170: As written, it sounds as though the authors infer that the polygons have formed in the valley bottom since 1959. Is it possible that the polygons pre-date 1959 but did not have a visible surface expression until warming begin to melt the ice wedges and causes local ground

subsidence that made the polygons visible in imagery and in the topography? I had always assumed, perhaps incorrectly without reading more background, that ice wedge polygon development was a slower longer-term process than time for them to appear at this study site.

Do the authors see any evidence for the development of pipes through the polygon network. Elsewhere melting of ice wedges by flowing water beneath intact ground surfaces is not uncommon (Fortier et al 2007). Collapse of soil into these tunnels can lead to channel development that is very distinct from gullying and headcutting commonly observed in more temperate systems. Do the authors see any evidence of this at this study site?

In Fig ED2 The caption notes that gravel bed material show imbrication, it might be worth mentioning this in the main text to indicate evidence that the gravels were transported by flowing water rather than just being lag deposits from loss of fine sediments and ice.

I was not able to access the movie. The link provided still required a password to sign in.

Reviewer #3 (Remarks to the Author):

Notes with Nature Communications Review, Jan 2023

This study uses field observations of a small valley river system in the Canadian high Arctic to investigate what are the major controls on channel incision and morphological change. The study site is located in the continuous permafrost region, meaning that the entire landscape is (still) dominated by frozen substrate and a seasonally active topsoil layer only. Ongoing and future projections of rapid warming will provoke rapid change in these systems and documentation of change in landscape processes is important and timely.

This paper is original and of relevance for anyone interested in evolution of permafrost landscapes. This topic is important because of the permafrost carbon feedback system which allows for previously frozen carbon to possibly decompose and release carbon dioxide or methane upon modern enhanced thaw. How that process plays out across the landscape is not well understood, and hydrological controls and channel corridors are thought to be especially complex due to geomorphic feedbacks of sediment transport and bank erosion. In addition, fundamental process understanding as presented in this paper applies to much wider zones of landscapes across the Northern hemisphere during the Quaternary glacial and interglacial cycles.

Noteworthy results include the detailed geomorphic descriptions of channel networks in a unique high-latitude cold region setting. These areas are extremely data sparse, and this study provides high-quality modern topographical data. In addition, analysis over 60+ years documents how sedimentation patterns are reshaping the topographic profiles under the present-day warming.

The study recognizes that well-developed channels are common around the edges of large polygons and within inter-connected troughs. But the data also indicates that channelization is diffuse or non-existent where relatively poorly developed polygons occur in wetlands. Thus, the paper provides as a finding how presence or absence of permafrost polygon geometry can influence or even control the channel pathways. This control has been hypothesized by other papers, and the authors do reference appropriate previous work. One exception: please reference to the work on polygon to hydrological connections by Liljedahl et al., <https://www.nature.com/articles/ngeo2674>, which appears perhaps more relevant than some of the work on Martian features.

The analysis of the topographic data also notes remarkable slope independence of the channelization process and demonstrates that W/D ratios do not necessarily follow erosive power or valley slope.

One major recommendation for improvement is to place these W/D ratios in perspective. The finding can be compared to lower latitudinal settings from literature, and to other Arctic settings.

The Arctic DEM data will allow for this analysis. I think such analysis deserves a figure.

Then, the authors ask what processes or factors control the production and discharge of water with a sufficient flow rate to help understand channelization within Muskox Valley? This the novel part of the manuscript!

Quantitative modeling of sediment transport processes and substrate thaw is invoked to provide understanding of entrainment of coarse sediment for frozen environments. This analysis is a new contribution, and my second major recommendation is to present this modeling more upfront in the paper. The model suggests that to transport frozen gravel, stream or river water needs to add heat first to thaw out the gravel from its matrix, and provides estimates of the time that thaw process would take. If stream discharge is cold and episodic, the duration of thawing out the bed may have inhibited transport even at hydraulically sufficient conditions. The description of these model results and Figure ED3 suggest that in frozen environments critical threshold sediment transport theory needs to be adapted. This can be better integrated into the main manuscript.

This study discusses implications of observations and available data, and provides suggestions for mechanisms based on field observations, high resolution topographic data and historical imagery and climate change, there is no direct evidence for some of the mechanisms. But this is typical for studies in such data sparse regions, the authors are careful to discuss the limited data, and the quantitative modeling further supports suggested controls.

Work is well documented in methods section. I applaud the authors for their intent to share the analysis as open-source workflows and for provided jupyter notebooks. The supplementary movies, figshare and jupyter notebooks are not viewable at this stage, even using the private links provided. So, I cannot evaluate this question properly and leave this question to the editor to verify.

I annotated the PDF with more minor suggestions to reword the abstract slightly, to make minor edits to figures etc.

Irina Overeem
University of Colorado

Response to Comments for Chartrand et al – NCOMMS-22-53478-T**Title: Interplay among climate, polygonal terrain and channel formation in the Canadian High Arctic****May 11, 2023****Response to Reviewer 1 Comments**Reviewer 1 Introductory Remarks:

I have reviewed the paper entitled ‘Interplay among climate, polygonal terrain and channel formation in the Canadian High Arctic’. This study tries to reveal the interactions among climate change, hydrologic processes governing water delivery, and consequently the timing and pace of channelization and landscape change in a high Arctic permafrost valley setting. Personally, I think this is an important contribution to the existing literature as there is a critical knowledge gap in terms of measuring the sedimentary and geomorphic responses to modern climate change (East et al., 2022, Earth's Future), as well as the possible impacts of rapid erosion on carbon release and thus climate feedbacks (Zhang et al., 2022, Nat. Rev. Earth Environ.). The findings of this study would be of interest to a range of research communities including climatologists, geomorphologists, as well as hydrologists and permafrost scientists. Although some of the statements in this study seem speculative and lack more robust in-situ observations (details below), it will likely inspire and stimulate more follow-up studies to enhance observations in the Arctic and develop new generations of more rigorous physical models of channel network development in the strategically important cold regions.

Overall, I think the manuscript is likely suitable for publishing in Nature Communications, after clarification of the major questions and several aspects listed below.

Response: Thanks for your constructive and supportive feedback. We appreciate your time and energy in reviewing and thinking about our work. Your comments have helped us prepare a more clear and impactful communication of our work, including development of new analysis (hydrodynamic modelling) and expansion of the thaw front calculations to highlight a particular knowledge gap we are presently pursuing.

Reviewer 1 Substantial Comments:

(1) The study seems lack quantitative findings and some statements are vague. For example, in Lines 24-26: this sentence seems the key message of this study, but the readers are left unclear about the specific findings. e.g., what are the specific major controls over channel incision in permafrost landscapes? what are the plausible positive feedbacks among the climate, periglacial landforms and channelization processes? Lines 26-29: again, it is not clear here and seems overstated. What is the new generic landscape change model (it is still unclear after reading throughout the paper)? what does the ‘intensifying variability’ represent? Do you mean the 'intensifying variability of climate, runoff, or sediment supply'?

Response: Thanks for pointing out the need for clarifications in the summary paragraph. We have addressed these comments in several ways:

1. At lines 24-26 of the original submission, we added the following phrase: “related to a proposed lake outburst flood event.”
2. At lines 26-29 of the original submission, we added the following phrase: “such as enhanced rates of permafrost degradation.”
3. We have also modified the last sentence of the summary paragraph as follows: “We integrate our observations and propose a new generic landscape change framework to better understand ~~model for the~~ underlying processes governing the channelization response...”
4. The “intensifying variability” phrase is with respect to the warming climate, as the sentence intends. No change made here.

(2) Introduction (Lines 35-55): it does summarize the existing knowledge gaps in a concise and comprehensive way and does highlight the importance of the seasonal dynamics of climate, ice, the active layer, different water sources and sediment supply in the permafrost environment in driving landscape change and channel initiation, but

this study lacks quantitative seasonal field observations of these key factors to support the underlying process. In particular, the seasonal water dynamics (multiple water sources in cold regions) and maybe also sediment supply can exert important controls on channel initiation and development. Maybe the data from similar permafrost landscapes would be helpful to explain some of the statements.

Response: To date, high elevation, Arctic and High Arctic research which notes and discusses channelization in broad terms does so in the context of suspended, or fine sediment. This is understandable because relatively fine sediment loads and dynamics are used as a proxy for watershed scale erosion conditions, and fine sediment is easier to measure and quantify compared to coarse, or bedload sediment. We see the the later problem as an emerging key to help advance landscape change science in these regions; however, we are unaware of any substantial records of bedload sediment measurement, particularly in High Arctic polar desert climates (we are aware of and have read Micheal Church's PhD research on Baffin Island – a wetter climate than the High Canadian Arctic). As a result, we have not attempted to incorporate data from other publications into the research we present. We have, however, addressed the comment with a few editorial changes, including the addition of the following sentence on line 39 of the original submission:

“The occurrence of active layer detachment slides and thaw slumps also influences hydrologic flow pathways by physically modifying the landscape, and providing point sources of meltwaters which can enhance down gradient flow channelization, introducing new sources of fine and coarse sediment to drainage basins^{9,19}.”

And we have added additional references at the end of two sentences within the range of line numbers of the original submission noted in the comment. We were remiss in not citing the noted publications in our original submission, and focused our referencing to research which relates directly to water sources known to increase fine sediment flux in permafrost landscapes, and as a result an apparent increase in watershed scale erosion.

(3) The study concludes that it “offers a local framework to develop new generations of rigorous physical models of valley-scale channel network development.” However, the readers are left unclear about the framework and also the process-based positive feedback with climate warming. Could you please explicitly explain process-based positive feedback by inserting a conceptual framework in Figure 4? The current Figure 4 is nice and helpful, and it can be improved by highlighting the interactions of different processes.

Response: Thanks for this important comment. We have carefully expanded Figure 4 (Figure 5 in the revised manuscript) to address the comment, while also being mindful that additional content could make the figure difficult to understand and therefore mask or hide the intent to focus on aspects of permafrost and periglacial landscapes which collectively contribute to drainage basin establishment and channelization. The principal additions to Figure 4 include:

1. Addition of a flow hydrograph, sediment flux and a drainage channel from a hypothetical glacier located at the left, mid-watershed part of the depicted basin. This links, in particular, to important research by Li et al. (2021) and Zhang et al. (2022) which highlights that glacier melt due to climate change (and downstream induced streamflow driven channel margin erosion due to thermal and mechanical effects) increases downstream fine sediment fluxes (and coarse sediment depending on the grain size distribution of eroded deposits and regolith).
2. Addition of a tributary channel originating from a thaw detachment slide (vignette G), that causes a fan at the valley margin, and a channel which then connects to the main channel at the valley center. This represents research by Walvoord et al. (2016), and Lafrenière and Lamoureux (2019) which highlights the importance of thaw slumps to basin sediment budgets. We also observed during our fieldwork that a large thaw slump located on a plateau above the valley floor resulted in the development of a new tributary channel which connected down to the valley trunk channel system  providing a direct input of seasonal water and sediments to the receiving basin.
3. Addition of two new features to vignette D: (a) a second permafrost boundary line (dashed, thinner line above the permafrost boundary line) which indicates that as the adjacent channel incises and deepens, the local permafrost boundary/active layer thickness increases and deepens, (b) a zoomed in snippet which shows a packed sediment bed with organics within the interstices of the sediments and which are subject to decomposition by microbes as the permafrost boundary deepens and the sediments seasonally dry. The

Decomposition and carbon respiration is dependent on the occurrence of buried organic carbon, and whether sediments/soil moisture content (e.g. Natali et al., 2015).

4. Addition of colorized relative sediment supply size distribution keys to various vignettes in the figure to indicate respective roles in delivery of sediment to the basin from the depicted processes, and as a result an overall/general indication of how each process is important for drainage basin establishment. Relative sediment supply is shown as being fine (suspended transport), or coarse (bedload transport). This is important because alluvial channels in mountainous landscapes are shaped predominately by bedload transport, but can be initiated through erosion that is predominately within fine, coarse or a size mixture of sediments. Several publications (referenced in the figure caption) were consulted in developing the colorized keys. The addition of the keys to the various vignettes (which previously focused on implications for water sourcing and movement through and within the landscape) and additional revisions discussed above provide a clearer idea of how different periglacial and permafrost landscape processes couple, or ‘interact’ to give rise to landscape development due to a warming climate.

Last, as we understand a ‘framework’ should (a) ask or motivate the development of impactful research questions; (b) provides an organizing structure for general understanding, and for research questions motivated by the framework, and (c) provides a general basis for interpreting how ‘things’ are understood (in the present case periglacial and permafrost landscapes responding to a warming climate), and as a basis for developing new hypotheses, etc. When we use ‘framework’ in the presentation this is what we mean; we believe the modifications made to Figure 4 based on the comments above helps in understanding the figure as a general framework.

(4) Similar to the High Mountain Asia (Li et al., 2021, Science), the Arctic climate has become warmer and wetter since the 1990s, with more summer rainfall that will likely change the hydrological processes and cause more permafrost disturbances and related erosion and channel network evolution. The changing rainfall patterns and their impacts on landscape changes (e.g., permafrost thermokarst development and channelization) should also be explicitly discussed (after a quick search, the word ‘rainfall’ only appears once in the main paper).

Response: Thanks for this important comment. We have incorporated a direct reference to rainfall at several additional locations throughout the main body of the manuscript:

1. Lines 151 of the Original Submission: “Over multiple seasons, presumably depending on the intensity of summertime warming ~~and~~ as well as rain and/or snowfall ~~precipitation~~ events⁴⁻⁶,...”;
2. Lines 211-212 of the Original Submission: “...subject to the magnitude of the seasonal water budget and the potential for proportionally more rainfall under a warmer Arctic³⁵ (Fig 5).” With this addition we have also incorporated an important new reference concerning future modelled precipitation trends in the Arctic (McCrystall et al., 2021, Nature Communications).

We also note that rainfall and its importance to the processes discussed was significantly incorporated within Figure 4 of the original submission.

Reviewer 1 Additional Comments:

(1) Line 79-81. This statement seems not supported by the data. Can you prove that the polygon geometry can influence or govern the channel path or paths both across and down topographic gradients at the valley scale?

Response: This was an important question which we considered carefully. We have completed additional analysis using a 2-dimensional hydrodynamic model to examine how polygons and the lack thereof influences surface flow routing along one part of Muskox Valley (what we call Panel C, of the LiDAR data). The simulations were conducted using BASEMENT-Basic Simulation Environment, ETH Zurich. The results are summarized in the new Figure 3, as well as supplemental material Figure ED1 and Table ED3. Figure 3 illustrates that polygons influence how surface flow is routed down valley for a surface flow of 2.5 cms, which is within the a range of flows that could have occurred during the hypothesized lake outburst flood (Table ED3). Supplemental Figure ED1 illustrates additional results from an plausible late summer, early fall flow (0.25 cms), to a an additional flow that could have occurred during the hypothesized lake outburst flood (5 cms) (Table ED3).

Furthermore, the distribution of downstream velocities differs from polygon regions to wetland regions, as expected. The map of surface flow routing in Figures 3 and ED1 also shows that water can move down the predominate topographic gradient, as well as across gradients as higher situated polygon troughs become flow paths for surface flow. In our original submission we inferred this behavior based on visual observations of similar flow behavior from our trip to Devon Island in the summer 2022; preparation of the model provides more concrete evidence of the inferred influence and represents a substantial addition to our overall presentation and supporting analysis. We note that the simulations were run for a fixed land surface boundary condition because we presently do not know how to model the entrainment of channel bed sediments which may be frozen, or partially frozen (as we discuss toward the end of the main body of the manuscript, and as elaborated by Zhang et al., 2022). All model set-up files have been uploaded to the manuscript data repository to facilitate reproduction by others.

(2) Line 137-139. Is it possible to provide more quantitative statements, such as the relative contributions of different water sources to emergent channel segments? How frequent is the outburst floods from the upstream lake?

Response: We presently lack the necessary data to make general statements about relative contributions of differing water sources to emergent channel segments, as well as how frequent outburst floods may be at our study site, or for similar landscapes. This is part of the focus of future research that we are seeking funding to complete. We believe the present sentence is true to what we do not know/understand, in the context of framing an important question.

(3) Line 287-290. It is not entirely clear how steeper hydraulic gradients provide increased surface areas for heat exchange.

Response: As a channel incises or deepens, there is more surface are exposed at the channel margins, provided the abandoned floodplain or adjacent surface changes elevation in a relatively minor way compared to the channel. We believe the source of confusion lies in how we worded the original sentence. We have modified the subject sentence as follows (additions in red text):

“Channel cross-section development within polygon troughs locally steepens hydraulic gradients, and a deepening channel provides increased surface areas at the channel margins for heat exchange, which can, in turn, increase rates of local permafrost thaw.”

(4) Movie: unfortunately, the Youtube link does NOT work and I did not see it from the manuscript tracking system.

Response: We apologize for the movie not being accessible. We have now made the movie public.

(5) Method-Calculation of Width-Depth Ratio. It is not easy to follow here. Can you provide a sketch diagram for the calculation of width and depth? For example, how do you determine the average depth and width for the cross-sections in Figure 2?

Response: Thanks for the clarifying comment/question. We have rewritten the section in the Field Site, Movies and Methods Summary section. The revised text is as follows:

“We acquired valley cross-section profiles on LiDAR elevation data to calculate width-depth ratios. We used ArcGIS 10.8.1 to clip the LiDAR data in four 600 m wide tiles along valley, measuring 1-2 km each. We then extracted 20 cross-sections spaced regularly every 70 m down valley, which were then loaded onto MATLAB to calculate cross-section slope and curvature (Fig. ED6). To minimize uncertainty related to qualitative measurements, we measured top width as the distance between curvature minima, and bottom width (valley floor) as the distance between curvature minima located within the previously identified valley width⁴⁵. Note that some profiles have multiple channels. These were also measured following the same technique. Associated depth along each transect was measured as the deepest point from the top width line.”

We have also prepared a new supplemental figure to illustrate an example of the cross-section slope and curvature (Figure ED6).

(6) Method-Calculation of the Mobility Analysis and Table ED2. I am not exactly working on incipient sediment motion. Here are a few questions for your consideration. Is the Shield’s analysis suitable to estimate sediment

particle mobilization conditions here? how about other methods that are more suitable for pebble-cobble sized sediment particle mobilization? For Table ED2, the authors seem to treat the riverbed sediment as a uniform bed when calculating the grains' mobilization velocity for a certain D50 or D90. However, the actual channel is a non-uniform bed, and how to consider it? The study selects 4 sets of experimental calculations (Table ED2). Are they representative and can they truly reflect the critical flow conditions for riverbed surface particles? What about the different ice conditions that could affect sediment mobility?

Response: Thanks for the questions and thoughts. We have conducted the originally presented sediment mobility analysis using the most generally accepted approach in fluvial sediment transport research. The calculations make no assumption of sediment size uniformity; on the contrary, a mixture of sizes is explicitly assumed in the use of a D50 and D90, as these representative sizes are drawn from a distribution of sizes. The question of whether the Shield's analysis is suitable or not is a fair question; until new theory is developed to a stage for making similar calculations, the Shield's analysis is the best science can offer when site information is somewhat limited. Furthermore, we present the outcomes of the calculations as 'estimates', noting there is uncertainty in the presented approach. We presently lack the necessary theory to make estimates of sediment mobility for partially or completely frozen channel bedsthis is the exact emphasis of the knowledge gap we present in the manuscript, and the focus of laboratory experiments we are presently building (within the introductory paragraph and in discussion of original submission Figure 4—i.e. the conceptual framework).

(7) Figs. 1-3. Please add the scale bars in these images. Would it be good to highlight the geographical information on the figure instead of describing it in the captions (Fig. 1 and Fig. 3).

Response: We have a several different responses here:

- It is difficult to insert a scale bar into Figure 1 because these are a variety of photographs taken from different altitudes and perspectives. However, we inserted two approximate length scale indications within Figure 1 based on the GIS coverage and our LiDAR DEM.
- In our Figure 2 original submission the x- and y-axes are in units of meters, and we indicate stationing along the channel profile line in meters (200 m increments). Based on the comments, we have added a scale bar to Figure 2a.
- In our Figure 3 original submission the top image provides an indication of approximate length scale, which is consistent between images. We have not added anything to Figure 3 in our revisions based on this comment.

(8) Fig. 2. Could the fonts in this figure be a bit larger?

Response: Sorry about that. We have increased the font size of the inset figures x- and y-axes labels, and the row labels in the power spectra of topography inset image.

(9) Line 33: change 'mountain regions' to 'cold regions'?

Response: We inserted 'seasonally cold and' in front of 'mountainous regions'

(10) Fig. 4: nice figure. Maybe the long caption could be shortened a bit. I did not see the framework from this figure.

Response: Thank you for the nice comment. We have left the caption as presented in the original submission (with some minor edits and additions) so that potential readers can understand the figure without needing to refer to the main text. We have addressed the concern regarding the idea of a 'framework' above, in our response to the earlier comment.

Response to Reviewer 2 Comments:

Reviewer 2 Introductory Remarks:

The manuscript provides a detailed field study of an evolving channel system in the Canadian High Arctic. Based on these observations that manuscript puts forward conceptual model for the relative controls of fluvial and

thermal controls on erosion and channel development in permafrost settings characterized by ice wedge polygons. The paper is well written and the graphics are excellent. I think the study is of broad value and highlights future data needs in the field. I reviewed an earlier submission of this manuscript and believe the paper is acceptable for publication in its present form. I only have a few minor points.

Response: Thanks for your constructive and supportive feedback. We appreciate your time and energy in reviewing and thinking about our work. We have addressed each of your comments through revisions, or further explanation provided below.

Reviewer 2 General Remarks:

(1) Figure 2: It is stated that the cross-section elevations are normalized by the maximum elevation. The axis reports elevation in m, but ranges from 0-1, are all of the cross sections only 1 m in height or should they be unitless? As mentioned in the prior review, it would be helpful if the manuscript explained what is learned by the power spectra and how it relates to the valley and channel evolution.

Response: Thanks for catching that mistake—yes, the elevations in the figure are unitless and we have corrected this error. To answer your second question, in our original submission we provided the following text within the Field Site, Movies and Methods Summary, under the Power Spectra Estimation section:

“Power spectral analysis of the ground topography is useful because it characterizes the dominant scale(s) of the topographic ground structure(s), or physical patterns. In the present case the power spectra reveal that polygon fields along the valley floor have a similar dominant length scale measured both down and across the valley.”

And we have provided the additional sentence following the above sentence within the same section:

“Furthermore and depending on how polygons are spatially organized, this dominant length scale imparts a relative and consistent unit flow path length as surface flows incise apparently stochastically along interconnected polygon troughs.”

(2) Line 165-170: As written, it sounds as though the authors infer that the polygons have formed in the valley bottom since 1959. Is it possible that the polygons pre-date 1959 but did not have a visible surface expression until warming begin to melt the ice wedges and causes local ground subsidence that made the polygons visible in imagery and in the topography? I had always assumed, perhaps incorrectly without reading more background, that ice wedge polygon development was a slower longer-term process than time for them to appear at this study site.

Response: This is an important question and comment—thanks. Our discussion is based on visual interpretation of the 1959 imagery, which could be prone to error. However, a careful review of the imagery suggests the occurrence of at least one set of polygons located roughly 350 m downstream of the lake at the head of Muskox Valley. Furthermore, review of Figure ED4 indicates that polygons were widespread at different locations within and near to Muskox Valley at the time of the 1959 imagery. As a result, we infer that other areas interpreted to have developed polygons at a later time, did not have polygons at the time of the 1959 imagery, or polygons were in the very early stages of development. Furthermore, if one assumes that polygons were widespread in Muskox Valley in 1959, it would be necessary to explain why polygons in some parts of the valley are much more prevalent or obvious in the imagery than in other areas where they might be assumed to occur, but cannot be visually inferred. We believe this is a more difficult position to maintain. Similar conclusions as proposed in the manuscript can be reached by comparing Figure ED4 with recent aerial imagery of Muskox Valley. As implied in your comment, this raises the issue that existing explanations of ice-wedge polygon development are incomplete, or are missing aspects that may introduce sensitivity to accelerated warming effects which can account for the relatively rapid development we suggest. This is part of future research we plan to address.

(3) Do the authors see any evidence for the development or pipes through the polygon network. Elsewhere melting of ice wedges by flowing water beneath intact ground surfaces is not uncommon (Fortier et al 2007). Collapse of soil into these tunnels can lead to channel development that is very distinct from gullying and headcutting commonly observed in more temperate systems. Do the authors see any evidence of this at this study site?

Response: This is a great question (one we asked ourselves many times), and one we evaluated carefully in the field, as well as this past summer when we completed fieldwork on Devon Island. To be very clear, no, we saw no

evidence of pipingnothing. As we mention in the original submission text, we cannot rule out that piping was/is not a part of the channelization process in Muskox Valley, but we see no present day evidence that it is active in the valley. We suspect that if it was active early on in the channelization process, most or all evidence was destroyed with the hypothetical lake outburst flood event(s). But we have no evidence for this so we do not raise this point in the manuscript. This is a major part of what makes the Muskox problem so intriguing.

The other aspect which we do not raise but hope to address with future experiments is the idea that piping can still be active, but obscured by coarse material that fills introduced voids/pipes on a time scale that basically matches the time scale of piping. In valley deposits rich with coarse sediments we hypothesize this is an alternate explanation for the development of channels, but with the apparent absence of pipes and other voids related to ice wedge melting by surface flow.

(4) In Fig ED2 The caption notes that gravel bed material show imbrication, it might be worth mentioning this in the main text to indicate evidence that the gravels were transported by flowing water rather than just being lag deposits from loss of fine sediments and ice.

Response: Thanks for this suggestion. We have modified the sentence at line 185 of the original submission to include reference to imbrication and Fig. ED2:

“Assuming a single outburst flood occurred, simple calculations on the basis of the approximate transport properties of sediment particles associated with plausible flow geometries suggests that the flood was capable of transporting gravel and cobble sized sediment^{30,31}, resulting in gravel and cobble imbrication observed at various locations along the channelized reaches (Methods; Figs. ED2 and ED3).”

(5) I was not able to access the movie. The link provided still required a password to sign in.

Response: We apologize for the movie not being accessible. We have now made the movie public.

Response to Reviewer 3 Comments:

Reviewer 3 Introductory Remarks:

This study uses field observations of a small valley river system in the Canadian high Arctic to investigate what are the major controls on channel incision and morphological change. The study site is located in the continuous permafrost region, meaning that the entire landscape is (still) dominated by frozen substrate and a seasonally active topsoil layer only. Ongoing and future projections of rapid warming will provoke rapid change in these systems and documentation of change in landscape processes is important and timely.

This paper is original and of relevance for anyone interested in evolution of permafrost landscapes. This topic is important because of the permafrost carbon feedback system which allows for previously frozen carbon to possibly decompose and release carbon dioxide or methane upon modern enhanced thaw. How that process plays out across the landscape is not well understood, and hydrological controls and channel corridors are thought to be especially complex due to geomorphic feedbacks of sediment transport and bank erosion. In addition, fundamental process understanding as presented in this paper applies to much wider zones of landscapes across the Northern hemisphere during the Quaternary glacial and interglacial cycles.

Noteworthy results include the detailed geomorphic descriptions of channel networks in a unique high-latitude cold region setting. These areas are extremely data sparse, and this study provides high-quality modern topographical data. In addition, analysis over 60+ years documents how sedimentation patterns are reshaping the topographic profiles under the present-day warming.

Response: Thank you for the supportive and constructive review. We appreciate the time and energy you put into thinking about the presented research.

Reviewer 3 General Remarks:

(1) The study recognizes that well-developed channels are common around the edges of large polygons and within inter-connected troughs. But the data also indicates that channelization is diffuse or non-existent where relatively

poorly developed polygons occur in wetlands. Thus, the paper provides as a finding how presence or absence of permafrost polygon geometry can influence or even control the channel pathways. This control has been hypothesized by other papers, and the authors do reference appropriate previous work. One exception: please reference to the work on polygon to hydrological connections by Liljedahl et al., <https://www.nature.com/articles/ngeo2674>, which appears perhaps more relevant than some of the work on Martian features.

Response: Thanks for pointing us to this publication and research. We were unaware of it, and agree on the relevance to Muskox Valley and implications for our presented research. We have incorporated it at relevant locations throughout the manuscript.

(2) The analysis of the topographic data also notes remarkable slope independence of the channelization process and demonstrates that W/D ratios do not necessarily follow erosive power or valley slope. One major recommendation for improvement is to place these W/D ratios in perspective. The finding can be compared to lower latitudinal settings from literature, and to other Arctic settings. The Arctic DEM data will allow for this analysis. I think such analysis deserves a figure.

Response: Thanks for this suggestion. We have brought in a basic comparison within the main body of the manuscript to alluvial rivers from the Pacific Northwest (large dataset), and from arid to semi-arid regions of the U.S. southwest (at line 122 of the original text submission). As expected, the width/depth from Muskox Valley falls within the ranges from the Pacific Northwest and the U.S. southwest. The exception being braided river segments where width/depth are commonly > 100 . We have not developed a new figure, or completed a pan-Arctic analysis of width-depth ratios. We believe this would be best as the focus of a different contribution. With this said, we do agree that the apparent independence between valley steepness and local width/depth is interesting and confusing—we hope to address this issue with theory development which will explore how systems like Muskox Valley evolve and respond to a warming climate.

(3) Then, the authors ask what processes or factors control the production and discharge of water with a sufficient flow rate to help understand channelization within Muskox Valley? This the novel part of the manuscript! Quantitative modeling of sediment transport processes and substrate thaw is invoked to provide understanding of entrainment of coarse sediment for frozen environments. This analysis is a new contribution, and my second major recommendation is to present this modeling more upfront in the paper. The model suggests that to transport frozen gravel, stream or river water needs to add heat first to thaw out the gravel from its matrix, and provides estimates of the time that thaw process would take. If stream discharge is cold and episodic, the duration of thawing out the bed may have inhibited transport even at hydraulically sufficient conditions. The description of these model results and Figure ED3 suggest that in frozen environments critical threshold sediment transport theory needs to be adapted. This can be better integrated into the main manuscript.

Response: Thanks for this important suggestion. We have addressed your suggestion in several ways:

1. Revised thaw front analysis: We revised the simple thaw front analysis to include a constant surface temperature forcing, interrupted by a hypothetical ‘heat wave’ at the surface, or heat transient. This revision provides a realistic set of circumstances in Arctic environments for the present warming regime. The main insight from this revised analysis is two-fold. (a) the thermal wave due to surface heat transients will likely take on the order of days to reach thaw front depths set by the background conditions, at which point thermal wave propagation will accelerate and penetrate more deeply (depending on local substrate and ground ice conditions). (b) the effects of a heat transient will be carried forward to the following thaw season because a deeper thaw front means more soil/substrate volume to freeze, subject to water availability, etc. This can likely have the effect of deepening the active layer over time, and drying out the landscape provided the seasonal precipitation regime is steady—this same effect was raised by Liljedahl et al. (2016), but due to differing processes related to polygon trough degradation. This highlights that multiple processes can interact or are reinforced and led to similar outcomes. We have brought this analysis into the main body of the manuscript including the revised figure. We also completed the revised analysis using the thaw front relationship reported by Kurylyk and Hayashi (2016) because it provides a

more direct way to represent the effects related to surface temperature events, and includes a simple correction factor for sensible heat effects.

2. As discussed above, we also developed basic flow routing simulations using the 2-dimensional hydrodynamic model Basement (ETH Zurich). We completed the modelling to evaluate how polygons, and the lack thereof, influence the routing of surface flows down topographic gradients. Furthermore, we wanted to provide evidence for the direct coupling between polygons and surface flows (which we observed in person during the summer 2022 on Devon Island—see new Movie detailed in the Field Site, Movie and Methods Summary) to support the proposal that polygons influence and likely accelerate rates of channelization where polygon troughs are interconnected and surface flows are large enough to mobilize channel bed sediments. As you point out in your comment, the next step is to develop a theoretical basis for sediment mobilization from partially or completely frozen channel beds—the seasonal timing of thawing such beds holds significant implications for sediment transport and channel evolution. These model simulations and implications have also been moved and incorporated into the main body of the manuscript (new Figure 3).

(4) This study discusses implications of observations and available data, and provides suggestions for mechanisms based on field observations, high resolution topographic data and historical imagery and climate change, there is no direct evidence for some of the mechanisms. But this is typical for studies in such data sparse regions, the authors are careful to discuss the limited data, and the quantitative modeling further supports suggested controls.

Response: Thanks for this supportive and rationale comment. We hope that experiments we are presently planning and building, as well as future planned trips to the High Arctic will shed additional light on some of the hypotheses we raise.

(5) Work is well documented in methods section. I applaud the authors for their intent to share the analysis as open-source workflows and for provided jupyter notebooks. The supplementary movies, figshare and jupyter notebooks are not viewable at this stage, even using the private links provided. So, I cannot evaluate this question properly and leave this question to the editor to verify.

Response: Apologies for the challenges in accessing the online repository of information. We have now corrected the issues and all supporting online material to promote open science is accessible.

(6) I annotated the PDF with more minor suggestions to reword the abstract slightly, to make minor edits to figures etc.

Response: Thank you for taking the time to annotate the original submission. Here are our responses:

1. Embedded Comment 1: We have left the phrase “This feature of” in the second sentence of the abstract. We use this phrase to refer to the amplification presented in the opening sentence. No changes made.
2. Embedded Comment 2: We believe the sentence reads okay, and captures the message we intend. No changes made.
3. Embedded Comment 3: Based on your suggestion, we have added the word “rising”. Thanks.
4. Embedded Comment 4: We noted the site is located in the Qikiqtani Region of NU in the opening sentence of the section. No changes made.
5. Embedded Comment 5: We do mention the power spectra in the original submission lines 98-103. We have also added one sentence to the Field Site, Movie and Methods Summary to better explain what we learn from the power spectra, based on a comment raised by Reviewer 2.
6. Embedded Comment 6: Thanks for this suggestion. We have added colored lines indicating the location of cross-sections shown in Figure 2a.
7. Embedded Comment 7: Thanks for catching that typo. Corrected to ‘later’ time.

8. Embedded Comment 8: Thanks. Noted and addressed above.
9. Embedded Comment 9: Thanks and good catch. We have deleted the power ‘-1’ from the annual precipitation value for Eureka, NU.
10. Embedded Comment 10: We addressed this comment above in relation to your width/depth comment.
11. Embedded Comment 11: Thanks.
12. Embedded Comment 12: We agree that there are other possible explanations for the loss of lake volume. Evaporation over a 60-year time frame seems unlikely with seasonal inputs of water each year in the 60-year period. We agree slow leakage is plausible. We have added the following sentences to the subject paragraph based on your comment:

“Is it also possible that all or a proportion of this volume was lost due to slow leakage at the downstream lake margin. Here we focus on the lake outburst loss mechanism as it provides the only plausible explanation for the balance of our field and photographic observations and inferences—in particular the imbrication of gravels and cobbles along channelized segments.”
13. Embedded Comment 13: Good point. However, given the lack of basis to calculate mobility for partially or completely frozen bed sediments, use of the Shield’s estimate is the best available alternative. We have prefaced the sentence with the following introductory phrase:

“Despite the uncertainty associated with the thermal state of bed sediments when the proposed flood may have occurred, we use a simple Shield’s analysis^{30,31} to estimate conditions necessary to mobilize gravel and cobble size sediment particles observed on the streambed in Muskox Valley for flow depths of 1 and 2 m (Figs. ED2 and ED3).”
14. Embedded Comment 14: Correct, we were unable to complete pebble counts. As noted above we have modified the text to indicate that the source of our grain size values is from visual and photographic estimates.
15. Embedded Comment 15: Thanks. As discussed above and in response to another comment you raised, we have incorporated this work within the main body of the revised manuscript.

END OF COMMENTS

References:

- Kurylyk, B. L., & Hayashi, M. Improved Stefan Equation Correction Factors to Accommodate Sensible Heat Storage during Soil Freezing or Thawing. *Permafrost and Periglacial Processes*, 27(2), 189–203 (2016).
- Lafrenière, M. J. and Lamoureux, S. F.: Effects of changing permafrost conditions on hydrological processes and fluvial fluxes, *Earth-Science Rev.*, 191, 212–223 (2019).
- Li, D., Overeem, I., Kettner, A. J., Zhou, Y., & Xixi, L. Air temperature regulates erodible landscape, water and sediment fluxes in the permafrost-dominated catchment on the Tibetan Plateau. *Water Resources Research* (2021).
- Liljedahl, A. K., Boike, J., Daanen, R. P., Fedorov, A. N., Frost, G. V., Grosse, G., et al. Pan-Arctic ice-wedge degradation in warming permafrost and its influence on tundra hydrology. *Nature Geoscience*, 9(4), 312–318 (2016).
- Natali, S. M., Schuur, E. A. G., Mauritz, M., Schade, J. D., Celis, G., Crummer, K. G., Johnston, C., Krapek, J., Pegoraro, E., Salmon, V. G., and Webb, E. E. Permafrost thaw and soil moisture driving CO₂ and CH₄ release from upland tundra. *J. Geophys. Res. Biogeosci.*, 120, 525– 537 (2015).
- McCrystall, M. R., Stroeve, J., Serreze, M., Forbes, B. C., & Screen, J. A. New climate models reveal faster and larger increases in Arctic precipitation than previously projected. *Nature Communications*, 12(1), 6765 (2021).
- Walvoord, M. A. and Kurylyk, B. L.: Hydrologic Impacts of Thawing Permafrost—A Review, *Vadose Zo. J.*, 15(6) (2016).
- Zhang, T., Li, D., East, A. E., Walling, D. E., Lane, S., Overeem, I., et al.-Warming-driven erosion and sediment transport in cold regions. *Nature Reviews Earth & Environment* (2022).

REVIEWERS' COMMENTS

Reviewer #1 (Remarks to the Author):

Thank you for the opportunity to review this revised manuscript. The authors have done well to comprehensively address my previous concerns and comments on the manuscript, and I sincerely thank them for their efforts in that regard. In particular, the new analysis on the 2-d hydrodynamic model enhances the confidence of the study. The manuscript is, in my opinion, now acceptable for publication. I believe this paper has broad interests to the community of permafrost/hydrology/geomorphology and will stimulate more follow-up studies and funds.

My last comment is that in the beginning of the study the authors could also cite East et al., 2022, *Earth's Future*; the idea is to place this paper in a larger context and bigger framework of measuring the sedimentary and geomorphic responses to modern climate change. In the discussion, the authors could further enhance the comparison with other cold regions to expand its broad readership.

East, A. E., Warrick, J. A., Li, D., Sankey, J. B., Redsteer, M. H., Gibbs, A. E.,... Barnard, P. L. (2022). Measuring and Attributing Sedimentary and Geomorphic Responses to Modern Climate Change: Challenges and Opportunities. *Earth's Future*, 10(10), e2022E-e2983E.

Dongfeng Li, Beijing

Reviewer #3 (Remarks to the Author):

This paper on "Interplay among climate, polygonal terrain and channel formation in the Canadian High Arctic" has undergone significant revision and thoughtful responses to issues earlier raised by reviewers have improved the manuscript.

The overall objective is now better clarified; to better understand stream and channel incisions in permafrost environments.

This overarching goal is motivated by the fact that permafrost environments are rapidly changing and that there are geomorphic and carbon cycle implications that are not fully constrained. But I also I find the point the authors make based on their channel width/depth data analysis convincing: existing common hydraulic geometry relationships between W/D and either streampower or slope were not found in Muskox valley. So, as the authors point out, it is this absence of relationships suggests different processes being of importance in these permafrost environments.

In the earlier version of the manuscript the reader was left pondering: What then are the most important processes?

One possible control is the role of polygons on channel evolution. I value how the newly added BASEMENT simulation hydrodynamics simulation better illustrates the control of polygons on routing of water in the landscape at low flow (2.5 m³/s). Field observations also document thaw slumping and other abrupt thaw features. And simple but novel modeling hints at a control of freeze-thaw dynamics on early season sediment transport. However, there remains some tension between the objective of understanding the interactions between climate change, seasonal permafrost dynamics and evolving channel geometry, and the confounding catastrophic event of a lake outburst, likely of utmost importance for this specific's valley incision and the cause of major sediment transport, but somewhat unique for Muskox valley and operating at a very different time scale than the more seasonal thaw. These discussions leads to one of the major conclusions; channel evolution in the Arctic is complicated, and many factors play a role. The proposed generic landscape evolution model provides a starting point for future research to quantify these processes.

Thank you for adding several key references, this made the paper well embedded in current research on this topic.

Figures were good and illustrative to begin with and have still improved based on suggestions.

I used the newly provided/fixed links to access the online repository of information. I confirm supporting online material is accessible and complete.

Response to Comments for Chartrand et al – NCOMMS-22-53478-A**Title: Interplay among climate, polygonal terrain and channel formation in the Canadian High Arctic****July 18, 2023****Response to Reviewer 1 Comments**Reviewer 1 Introductory Remarks:

Thank you for the opportunity to review this revised manuscript. The authors have done well to comprehensively address my previous concerns and comments on the manuscript, and I sincerely thank them for their efforts in that regard. In particular, the new analysis on the 2-d hydrodynamic model enhances the confidence of the study. The manuscript is, in my opinion, now acceptable for publication. I believe this paper has broad interests to the community of permafrost/hydrology/geomorphology and will stimulate more follow-up studies and funds.

Response: Thanks for your constructive and supportive feedback. We appreciate your time and energy in reviewing and thinking about our work. Your comments have helped us prepare a more clear and impactful communication of our work.

Reviewer 1 Additional Comments:

My last comment is that in the beginning of the study the authors could also cite East et al., 2022, Earth's Future; the idea is to place this paper in a larger context and bigger framework of measuring the sedimentary and geomorphic responses to modern climate change. In the discussion, the authors could further enhance the comparison with other cold regions to expand its broad readership.

Response: Thanks for pointing out the East et al., 2022 publication. We are familiar with it, as well as the predecessor East and Sankey, 2020 publication. Both publications are important contributions to the interdisciplinary science of climate change and geomorphology. We believe, however, for the purposes of the present work, the East and Sankey (2020)¹ publication provides more direct connection and relevance to some of the issues we discuss, notwithstanding the fact that both publications are much more broad in scope than we attempt here. Furthermore, East and Sankey (2020) setup much of the issues discussed in the East et al. (2022) publication. As a result, we have integrated East and Sankey (2020) into our work here.

Specifically, we cite their research in relation to (1) the idea that environmental change is in general complex due to the numerous co-evolving environmental factors, and (2) in relation to the knowledge gap related to channelization and channel network development, which sits as a gap at the interface between climate change and geomorphology. This last point was a main issue raised by East and Sankey (2020) as a critical area that geoscientists should focus their research. The East and Sankey (2020) publication is the new reference #27 in the manuscript. No other changes made in relation to other cold regions. Our present manuscript cites and discusses research from Antarctica, other locations in the western hemisphere Arctic, the Siberian Arctic and the Tibetan plateau. We believe this is sufficient for the focus of our presented research.

Response to Reviewer 3 Comments:Reviewer 3 Introductory Remarks:

This paper on “Interplay among climate, polygonal terrain and channel formation in the Canadian High Arctic” has undergone significant revision and thoughtful responses to issues earlier raised by reviewers have improved the manuscript.

1. East, A. E., & Sankey, J. B. (2020). Geomorphic and Sedimentary Effects of Modern Climate Change: Current and Anticipated Future Conditions in the Western United States. *Reviews of Geophysics*, 58(4), e2019RG000692-e2019RG000692. <https://doi.org/10.1029/2019RG000692>.

The overall objective is now better clarified; to better understand stream and channel incisions in permafrost environments.

This overarching goal is motivated by the fact that permafrost environments are rapidly changing and that there are geomorphic and carbon cycle implications that are not fully constrained. But I also find the point the authors make based on their channel width/depth data analysis convincing: existing common hydraulic geometry relationships between W/D and either streampower or slope were not found in Muskox valley. So, as the authors point out, it is this absence of relationships suggests different processes being of importance in these permafrost environments.

In the earlier version of the manuscript the reader was left pondering: What then are the most important processes? One possible control is the role of polygons on channel evolution. I value how the newly added BASEMENT simulation hydrodynamics simulation better illustrates the control of polygons on routing of water in the landscape at low flow (2.5 m³/s). Field observations also document thaw slumping and other abrupt thaw features. And simple but novel modeling hints at a control of freeze-thaw dynamics on early season sediment transport. However, there remains some tension between the objective of understanding the interactions between climate change, seasonal permafrost dynamics and evolving channel geometry, and the confounding catastrophic event of a lake outburst, likely of utmost importance for this specific's valley incision and the cause of major sediment transport, but somewhat unique for Muskox valley and operating at a very different time scale than the more seasonal thaw.

These discussions leads to one of the major conclusions; channel evolution in the Arctic is complicated, and many factors play a role. The proposed generic landscape evolution model provides a starting point for future research to quantify these processes.

Thank you for adding several key references, this made the paper well embedded in current research on this topic.

Figures were good and illustrative to begin with and have still improved based on suggestions.

I used the newly provided/fixed links to access the online repository of information. I confirm supporting online material is accessible and complete.

Response: Thank you for the supportive and constructive review. We appreciate the time and energy you put into thinking about the presented research. We agree with you that future research should emphasize more concrete examination of “interactions between climate change, seasonal permafrost dynamics and evolving channel geometry”. We intend to focus on this issue, in part, with future numerical simulations and fieldwork. No changes made based on Reviewer 3 comments.